# 4D-LRM: Large Space-Time Reconstruction Model From and To Any View at Any Time

**Ziqiao Ma**[1,2]    **Xuweiyi Chen**[4]    **Shoubin Yu**[1,3]

**Sai Bi**[1]    **Kai Zhang**[1]    **Ziwen Chen**[1,5]    **Sihan Xu**[2]    **Jianing Yang**[1,2]

**Zexiang Xu**[1]    **Kalyan Sunkavalli**[1]    **Mohit Bansal**[3]    **Joyce Chai**[2]    **Hao Tan**[1]

[1]Adobe Research    [2]University of Michigan

[3]UNC Chapel Hill    [4]University of Virginia    [5]Oregon State University

https://4dlrm.github.io/

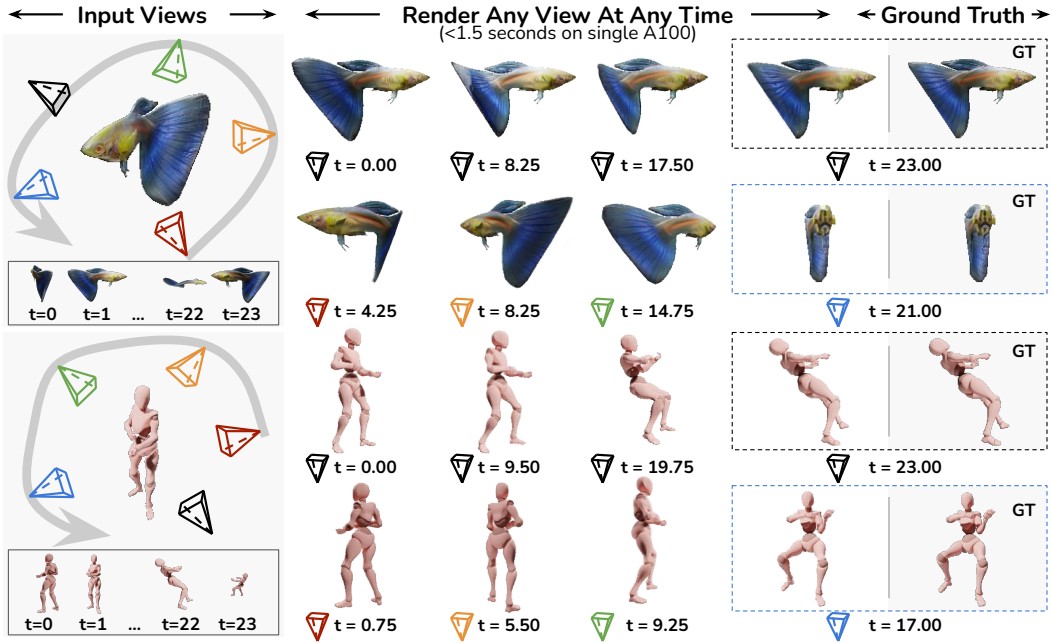

Figure 1: Large Space-Time Reconstruction Model (4D-LRM) is a data-driven 4D reconstruction model that takes sparse input views at any time and renders arbitrary novel view-time combinations.

## Abstract

Can we scale 4D pretraining to learn general space-time representations that reconstruct an object from a few views at some times to any view at any time? We provide an affirmative answer with 4D-LRM, the first large-scale 4D reconstruction model that takes input from unconstrained views and timestamps and renders arbitrary novel view-time combinations. Unlike prior 4D approaches, e.g., optimization-based, geometry-based, or generative, that struggle with efficiency, generalization, or faithfulness, 4D-LRM learns a unified space-time representation and directly predicts per-pixel 4D Gaussian primitives from posed image tokens across time, enabling fast, high-quality rendering at, in principle, infinite frame rate. Our results demonstrate that scaling spatiotemporal pretraining enables accurate and efficient 4D reconstruction. We show that 4D-LRM generalizes to novel objects, interpolates across time, and handles diverse camera setups. It reconstructs 24-frame sequences in one forward pass with less than 1.5 seconds on a single A100 GPU.

39th Conference on Neural Information Processing Systems (NeurIPS 2025).

# 1 Introduction

Reconstructing dynamic objects and scenes from video is a fundamental problem in computer vision full of challenges. The ability to accurately capture both spatial structure and temporal dynamics to build a complete 4D representation from limited visual inputs, across varying views and time, would significantly advance applications, such as 4D asset generation [78, 53] for video games, film production, and AR/VR, as well as world modeling [32, 96] for embodied AI and robotics.

Prior work on 4D modeling generally follows three directions, each shaped by different assumptions, target applications, or inherent limitations. The first direction is **optimization-based**. These methods reconstruct space and time by optimizing per scene or object from multi-view videos [3, 28, 82]. While these methods can produce high-quality results, they require dense spatial and temporal sampling, limiting their practicality with sparse inputs. The second direction is **geometry-based**. Motivated by Geometry Transformers [71, 66], they aim to estimate dynamic geometry and extract camera poses or depth maps directly from videos [90, 19, 70]. This line is orthogonal to the previous line of work, as these methods are not intended for novel view or time synthesis and instead focus on per-frame geometry estimation [90]. The third direction is **generation-based**, aiming to produce perceptually plausible 4D assets using visual generative models, particularly video diffusion models [73, 76, 78, 83]. These methods require fewer inputs, but are still computationally expensive, sensitive to prompts [36], or tailored for single-view monocular videos (Figure 2a) [53, 78]. As Yao et al. [83] noted, generating dynamic 3D content from a single-view video is inherently ill-posed for reconstruction due to motion ambiguity. Thus, these methods prioritize perceptual quality over faithful 4D reconstruction.

Recent advances in Large Reconstruction Models (LRMs) [24, 92, 98] offer a promising **rendering-based** alternative toward efficient and high-quality 3D reconstruction. Based on Transformer architectures, LRMs have shown strong performance on 3D reconstruction tasks by learning powerful priors over shape and appearance from large-scale 3D datasets. This also enables them to reconstruct detailed objects and scenes from only a few posed images. However, existing LRMs are designed for static 3D objects or scenes. Although recent work has explored adapting them for 4D asset generation [53] and scene-level reconstruction with limited camera dynamics [81, 39], extending LRMs to general 4D reconstruction remains challenging, particularly when dealing with sparse multi-views and missing timestamps. We envision that an ideal 4D reconstruction model should be able to learn accurate spatiotemporal representations from a limited set of input views at a few timestamps, enabling reconstruction at novel view-times by effectively sharing information across both viewpoint and time (Figure 2b). This motivates a fundamental question for 4D modeling:

*Can we scale 4D pretraining to learn a generic space-time representation that reconstructs an object from a few views at some time points, to any view at any time?*

We introduce 4D-LRM, a Transformer-based large space-time reconstruction model for dynamic object reconstruction, trained in a data-driven manner. Inspired by 4D Gaussian Splatting (4DGS; [82]), 4D-LRM adopts a unified treatment of space and time, representing a dynamic object as a cloud of anisotropic 4D Gaussians. As illustrated in Figure 3, we patchify temporally posed input images into image tokens, which are processed by Transformer blocks. The model then directly regresses per-view, per-pixel 4D Gaussian primitives from contextualized multi-view tokens across time. These predicted 4DGS primitives enable fast, high-quality reconstruction and rendering from any viewpoint at any timestamp, with, in principle, an infinite frame rate. We train 4D-LRM on a curated subset of Objaverse4D [13], consisting of dynamic, articulated objects captured over time. The model scales effectively with more data and larger model size, and generalizes well to novel objects.

To the best of our knowledge, 4D-LRM is the first large-scale 4D reconstruction model that supports input from unconstrained posed views and timestamps, and renders arbitrary novel view-time combinations. With around 300M parameters, 4D-LRM achieves a high-quality reconstruction on Consistent4D [28] and the hold-out test set of Objaverse4D using only one input view per frame. It reconstructs a 24-frame dynamic object in one forward pass with less than 1.5 seconds on a single A100 GPU. Compared to per frame 3D reconstruction, 4D-LRM exhibits strong and robust performance under diverse input camera configurations. We attribute this to 4D-LRM's ability to jointly model spatial and temporal contexts, effectively resolving motion ambiguities by sharing information across views and time. We further unlock its application to 4D asset generation, which surpasses baselines in both faithfulness and inference speed. Finally, we present detailed scaling behavior analyses for both training and inference, examining the scalability of different design choices and how inference

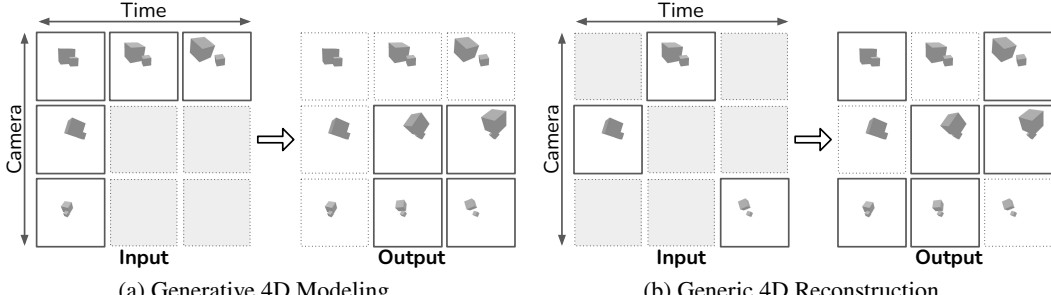

(a) Generative 4D Modeling.  (b) Generic 4D Reconstruction.

Figure 2: Comparison between previous generative 4D modeling methods (e.g., L4GM [53] and SV4D [78, 83]) and our goal of generic 4D reconstruction. Prior approaches take a single monocular video as input and use generative priors to synthesize multi-view images for the first frame. In contrast, our objective is to reconstruct dynamic objects from any viewpoint at any timestamp.

performance varies with the number of input views. This highlights future directions in developing 4D-LRM variants that can handle longer contexts [98] and support test-time training [11].

## 2 Large Space-Time Reconstruction Model (4D-LRM)

### 2.1 Preliminary: 4D Gaussian Splatting (4DGS)

**3D Gaussian Splatting.** With 3D Gaussian Splatting (GS; [31]), a static 3D scene can be represented as a cloud of anisotropic 3D Gaussians. Each Gaussian is in principle unbounded and, unless filtered, contributes to a point $x \in \mathbb{R}^3$ via an unnormalized density:

$$p(x|\mu, \Sigma) = \exp\left[-\frac{1}{2}(x - \mu)^T \Sigma^{-1}(x - \mu)\right], \tag{1}$$

where $\mu = (\mu_x, \mu_y, \mu_z) \in \mathbb{R}^3$ is the mean and $\Sigma \in \mathbb{R}^{3 \times 3}$ is the covariance. The covariance is factorized as $\Sigma = RSS^T R^T$, with $S = \text{diag}(s_x, s_y, s_z)$ and $R$ derived from a unit quaternion $q$.

**Pixel-Aligned Gaussians.** GS-based reconstruction models [6, 60, 92] adopt pixel-aligned Gaussian rendering, where the center of each 3D Gaussian is computed from the ray distance and known camera parameters. Given ray origin $\text{ray}_o$, direction $\text{ray}_d$, and distance $\delta$, the center is $\mu = \text{ray}_o + \delta \cdot \text{ray}_d$. Therefore, each Gaussian can be parameterized with $\dim_{\text{3DGS}} = 12$, including 3-channel RGB color, 3-channel scale, 4-channel rotation quaternion, 1-channel opacity, and 1-channel ray distance.

**4D Gaussian Splatting.** When considering a dynamic scene, Yang et al. [82] observed that treating space and time as independent, i.e., assuming $p_i(x, y, z|t) = p_i(x, y, z)$ for the $i$-th visible Gaussian, is undesirable. Instead, they extend the formulation of Kerbl et al. [31] to model dynamic scenes with a unified treatment of spatial and temporal dimensions using a coherent 4D Gaussian representation (4DGS; [82]). The mean of 4DGS is given by $\mu = (\mu_x, \mu_y, \mu_z, \mu_t) \in \mathbb{R}^4$, which captures both the spatial and temporal centers. 4DGS parameterizes its covariance matrix $\Sigma$ as a 4D ellipsoid $\Sigma = RSS^T R^T$, where $S = \text{diag}(s_x, s_y, s_z, s_t)$ is a diagonal scaling matrix and $R \in \mathbb{R}^{4 \times 4}$ is a 4D rotation matrix. In 4D Euclidean space, $R$ can be decomposed into a pair of left and right isoclinic rotations, each represented by a quaternion. Together, a general 4D Gaussian can be parameterized with $\dim_{\text{4DGS}} = 20$, including 3-channel RGB color, 4-channel scale, two 4-channel quaternions, 1-channel opacity, and the 4-channel space-time centers of order xyzt.

**Sampling Conditional 3DGS from 4DGS.** As is shown in Figure 3, the marginal probability $p_i(t)$ at time $t$ is a one-dimension Gaussian $\mathcal{N}(t; \mu_4, \Sigma_{4,4})$. The conditional 3DGS can be derived from the properties of the multivariate Gaussian with:

$$\begin{aligned}
\mu_{xyz|t} &= \mu_{1:3} + \Sigma_{1:3,4} \Sigma_{4,4}^{-1}(t - \mu_4), \\
\Sigma_{xyz|t} &= \Sigma_{1:3,1:3} - \Sigma_{1:3,4} \Sigma_{4,4}^{-1} \Sigma_{4,1:3}.
\end{aligned} \tag{2}$$

This decomposition enables direct adaptation of the 3DGS tile-based rasterizer by first evaluating the marginal distribution $p_i(t)$, allowing for the accumulation of color and opacity over time. More details are available in Appendix B.2.

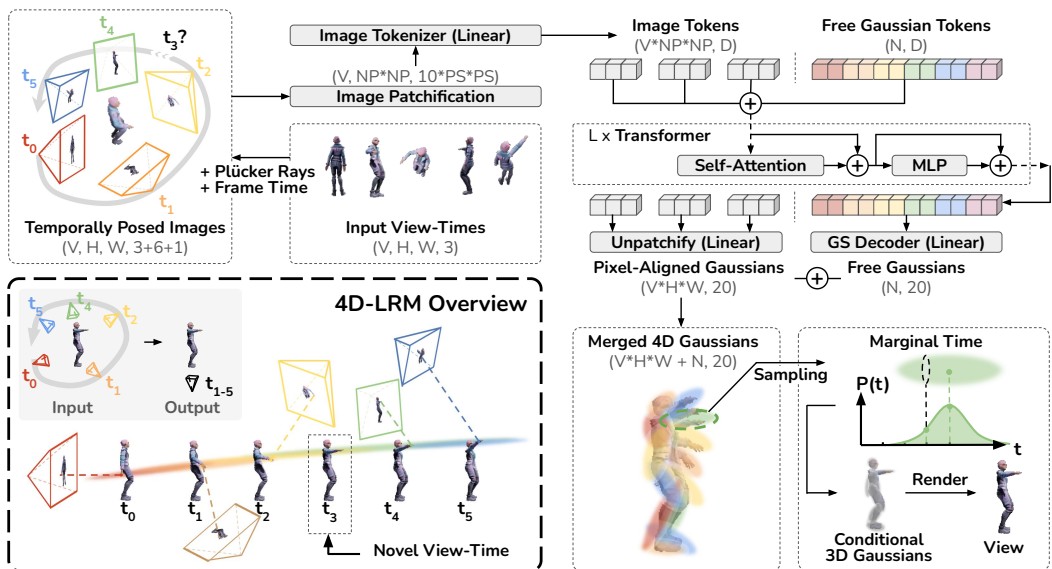

Figure 3: Overview of 4D-LRM. 4D-LRM adopts a unified treatment of space and time, representing a dynamic object as a cloud of anisotropic 4D Gaussians [82]. We train a simple Transformer to regress 4D Gaussian primitives from a set of images with camera poses and timestamps. Each input image is tokenized by patchifying the temporally posed frames. The resulting multi-view image tokens are concatenated in temporal order and passed through a series of transformer blocks. An optionalal set of $N$ learnable free Gaussian tokens append the image tokens for greater generative flexibility.

## 2.2 Transformer-Based Image-to-4DGS Decoder

**Tokenizing Temporally Posed Images.** As shown in Figure 3, the inputs to our model are $V$ images from arbitrary view-time combinations, denoted as $\{\mathbf{I}_j \in \mathbb{R}^{H \times W \times 3}\}$ for $j = 1, 2, .., V$, along with their corresponding camera intrinsic and extrinsic parameters. Here, $H$ and $W$ represent the image height and width, respectively. For pose conditioning, we compute Plücker ray coordinates [48] for each image, resulting in $\{\mathbf{P}_j \in \mathbb{R}^{H \times W \times 6}\}$. Instead of the standard canonical Plücker coordinates $[\mathrm{ray}_d, \mathrm{ray}_o \times \mathrm{ray}_d]$, we follow GS-LRM [92] and represent each ray as a direction plus its closest point to the origin $[\mathrm{ray}_d, \mathrm{ray}_o - \langle \mathrm{ray}_o, \mathrm{ray}_d \rangle \cdot \mathrm{ray}_d]$, which is suitable for pose-sensitive learning and pixel alignment. Temporal conditioning is provided by a timestamp map $\{\mathbf{T}_j \in \mathbb{R}^{H \times W \times 1}\}$. We channel-wise concatenate the RGB values, Plücker coordinates, and time to obtain a per-view feature map $\widetilde{\mathbf{I}}_j = \mathrm{Concat}(\mathbf{I}_j, \mathbf{P}_j, \mathbf{T}_j)$ of 10 channels, enabling per-pixel pose and time conditioning, naturally serving as spatial and temporal embeddings to distinguish different patches. Therefore, we do not use additional positional embeddings. Following Vision Transformer (ViT) architectures [16], we divide each per-view feature map into non-overlapping patches of size $PS^2$. Each patch is flattened into a vector embedding of size $10 \cdot PS^2$, and then projected to a token of dimension $D$ (the Transformer width) using a linear layer.

**Decoding Per-Pixel Gaussians with Transformer.** We concatenate the image tokens and feed them through $L$ layers of Transformer blocks. Each block follows a standard architecture consisting of Pre-LayerNorm [2], multi-head self-attention [64], and MLP, all equipped with residual connections [23]. The output tokens are then unpatchified and decoded into pixel-aligned 4D Gaussian primitives using a single linear layer. Given the same patch size, this results in $V \times H \times W$ Gaussians, each with $\dim_{4DGS} = 20$. As in prior GS-based LRMs, we adopt pixel-aligned Gaussian rendering. From each decoded 4D Gaussian parameter $\mathbf{g} \in \mathbb{R}^{20}$, we split the 4-channel space-time vector $(\mathbf{g}_x, \mathbf{g}_y, \mathbf{g}_z, \mathbf{g}_t)$, retain the time $\mu_t = \mathbf{g}_t$, and normalize the xyz features to a scalar distance $\delta$. Further details on 4DGS parameter initialization and differentiable rasterization are provided in Appendix B.

**Optional Free Gaussians.** Pixel-aligned Gaussians scale naturally with input resolution, making them well-suited for generalization to higher resolutions [92]. However, this design becomes suboptimal for very sparse views or setups with limited view coverage over motion, such as those used in standard generative 4D modeling [53, 78]. To address this, we optionally introduce an additional set of $N$

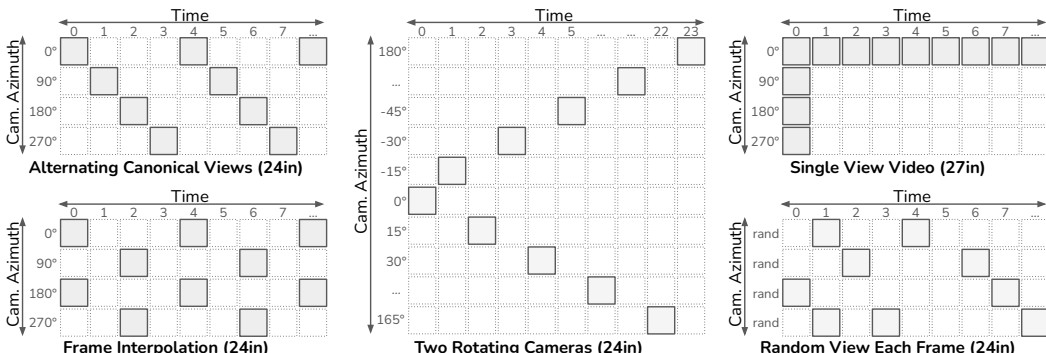

Figure 4: Different types of camera setups in evaluations: **Alternating Canonical Views** (4 camera poses, 24/24 timestamps seen, 24 input views in total); Frame Interpolation (4 camera poses, 12/24 timestamps seen, 24 input views in total); **Two Rotating Cameras** (24 camera poses, 24/24 timestamps seen, 24 input views in total); **Single View Video** [53, 78] (4 camera poses on the first frame plus a single view video for subsequent frames, 24/24 timestamps seen, 27 input views in total); **Random Input Views** (random poses, 24/24 timestamps seen, 24 input views in total).

learnable Gaussian tokens, concatenated with the image tokens. These tokens allow the model to generate freeform 4D Gaussian primitives. Unlike pixel-aligned Gaussians, these do not rely on pose or time conditioning. Instead, we use a separate linear layer to decode the 4-channel space-time vector $(\mathbf{g}_x, \mathbf{g}_y, \mathbf{g}_z, \mathbf{g}_t)$ that directly defines the Gaussian center $\mu = (\mu_x, \mu_y, \mu_z, \mu_t)$ after activation. We describe in Section 3.1 how 4D-LRM can be fine-tuned for 4D asset generation with free Gaussians.

## 2.3 Training Objectives

During training, we render images at $U$ supervision views using the predicted 4D Gaussians and minimize the image reconstruction loss. Let $\{\mathbf{I}^*_{i'} \mid i' = 1, 2, \ldots, U\}$ denote the ground truth views and $\{\widehat{\mathbf{I}}^*_{i'}\}$ the corresponding rendered images. The training loss combines Mean Squared Error (MSE) and Perceptual loss [9]:

$$\mathcal{L} = \frac{1}{U} \sum_{i'=1}^{U} \left( \text{MSE}(\widehat{\mathbf{I}}^*_{i'}, \mathbf{I}^*_{i'}) + \lambda \cdot \text{Perceptual}(\widehat{\mathbf{I}}^*_{i'}, \mathbf{I}^*_{i'}) \right), \tag{3}$$

where $\lambda$ controls the weight of the perceptual loss and is set to 0.5 empirically.

# 3 Experiments

## 3.1 Implementation Details

**Training Data.** To enable large-scale training, we construct a 4D dataset derived from Objaverse [13], which provides a subset of animated 3D assets. However, the raw dataset is not directly suitable for 4D modeling: object motions are often inconsistent, and the dataset contains duplicates and artifacts. To address this, we build upon the filtered subset curated by Diffusion4D [38], which removes static objects and unstable motion sequences, leading to 32,000 animated objects. For each object, we render a 24-frame video from diverse camera trajectories. We augment the dataset with 783,000 static 3D objects from Objaverse by treating each as a 24-frame video, applying minor frame-by-frame displacements along a single random direction. More details are available in Appendix B.3.

**Benchmark Data.** We use the Consistent4D [28] objects for evaluation. Additionally, we hold out 56 challenging objects exhibiting more complex motion as an extended test set to support future benchmarking. For evaluation, we re-render the first 48 ($2 \times 24$) frames of the Consistent4D dataset and the first 24 frames of the Objaverse4D (Test) split. During training, we exclude 6 (out of the 7) Consistent4D test objects that also appear in Objaverse from our training set to avoid data leakage.

**Curriculum Learning.** We adopt a curriculum learning strategy to reduce computational cost. Specifically, we pretrain the model at a resolution of $128 \times 128$ for 100,000 steps, and then continue at $256 \times 256$ for an additional 20,000 steps. The continual pretraining stage uses the same model architecture, initialized with the same pre-trained weights, but processes more tokens due to more

Table 1: Breakdown evaluation of each camera setup on Consistent4D (Re-rendered) dataset. For each setup, we evaluate and average the score on 4 canonical views and 1 randomly sampled view. We consider different **Res**olutions and model **Init**ialization strategies and compare to GS-LRM [92].

| Res. | Models | Init. | Alter. Canonical Views | | | Frame Interpolation | | | Two Rotating Cameras | | | Random Input Views | | |
|---|---|---|---|---|---|---|---|---|---|---|---|---|---|---|
| | | | PSNR | LPIPS | SSIM | PSNR | LPIPS | SSIM | PSNR | LPIPS | SSIM | PSNR | LPIPS | SSIM |
| 128 | 4D-LRM-Base | No | 29.233 | 0.047 | 0.961 | 29.194 | 0.047 | 0.961 | 25.014 | 0.071 | 0.926 | 25.788 | 0.081 | 0.926 |
| | 4D-LRM-Large | No | 30.274 | 0.038 | 0.969 | 30.260 | 0.038 | 0.969 | 25.904 | 0.061 | 0.935 | 26.525 | 0.070 | 0.934 |
| | 4D-LRM-Large | Yes | **31.023** | **0.031** | **0.974** | **30.917** | **0.031** | **0.973** | **28.703** | **0.042** | **0.959** | **28.789** | **0.049** | **0.957** |
| 256 | SoM [69] | - | 25.586 | 0.055 | 0.906 | - | - | - | 22.756 | 0.089 | 0.941 | 17.637 | 0.208 | 0.875 |
| | GS-LRM (Per Fr.) | - | 19.327 | 0.097 | 0.902 | - | - | - | 20.037 | 0.094 | 0.935 | 16.826 | 0.212 | 0.801 |
| | GS-LRM (All in) | - | 21.606 | 0.086 | 0.925 | 21.590 | 0.086 | 0.925 | 20.641 | 0.100 | 0.909 | 19.665 | 0.132 | 0.897 |
| | 4D-LRM-Base | No | 27.443 | 0.062 | 0.952 | 27.394 | 0.062 | 0.953 | 23.429 | 0.088 | 0.918 | 23.882 | 0.096 | 0.916 |
| | 4D-LRM-Large | No | 27.860 | 0.049 | 0.959 | 27.822 | 0.048 | 0.959 | 25.095 | 0.069 | 0.934 | 25.776 | 0.073 | 0.934 |
| | 4D-LRM-Free | Yes | 30.396 | 0.036 | 0.973 | 30.376 | 0.036 | 0.973 | 26.184 | 0.061 | 0.943 | 26.337 | 0.067 | 0.939 |
| | 4D-LRM-Large | Yes | **32.177** | **0.028** | **0.980** | **32.145** | **0.028** | **0.980** | **27.664** | **0.050** | **0.957** | **27.990** | **0.057** | **0.954** |

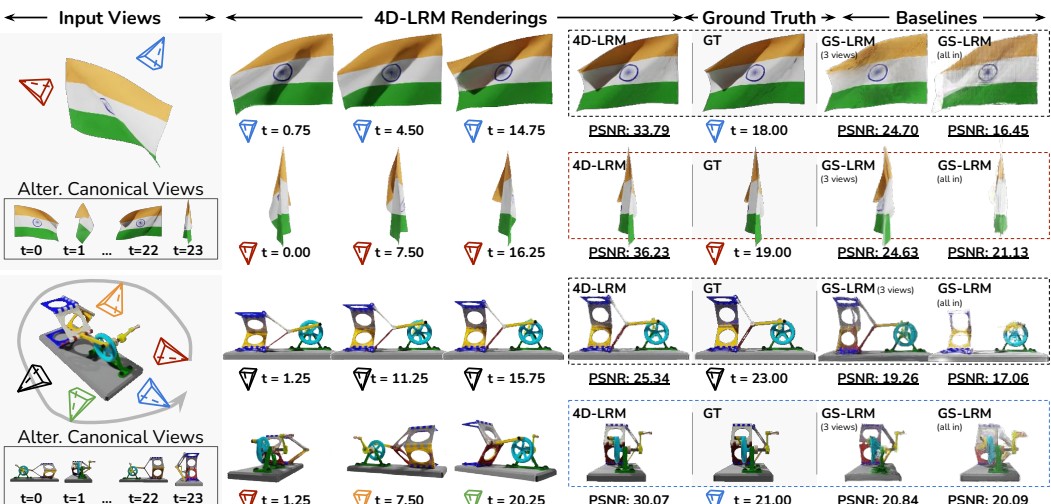

Figure 5: Visual comparison with GS-LRM [92] using (a) all input views across time and (b) three random views from the same timestamp. 4D-LRM reconstructs novel view-time combinations by learning spatiotemporal representations from sparse inputs, outperforming per-frame reconstruction by effectively sharing information across both space and time.

pixel-aligned Gaussians in higher resolution. At each training step for object-level data, we randomly sample 36 images (from 144 renderings over 24 frames) as a training example. From this set, we independently select 12 input views and 24 supervision views, allowing overlap to improve convergence. Both pretraining stages are performed on 160 A100 GPUs. The first stage adopted a per-GPU batch size of 16 and took approximately 5 days, while the second stage adopted a per-GPU batch size of 8 with an additional 5 days. Unless otherwise specified, most training configurations follow GS-LRM [92]. Additional implementation and training details are provided in Appendix B.4.

**Fine-tuning 4D-LRM for 4D Generation.** In the pre-training stage, we do not include any free Gaussians. We fine-tune 4D-LRM with $N = 4096$ free Gaussians for the 4D generation task similar to the setting of [53, 78]. For each training example over 24 frames, we select 4 canonical views at the initial frame and all views from a single-view monocular video as input. We randomly sample 8 supervision views. This is trained on 64 A100 GPUs with a per-GPU batch size of 8 for 16,000 steps.

## 3.2 Experiment Setups

**Input Camera Setup.** Since 4D-LRM supports arbitrary input views at any time, we define several camera setup configurations fore broad and systematic evaluation (see Figure 4 for illustrations):

• **Alternating Canonical Views.** The input view alternates cyclically among four canonical directions (front, left, back, and right) across frames;

Table 2: Breakdown evaluation of each camera setup on Objaverse4D (Test) dataset. For each setup, we evaluate and average the score on 4 canonical views and 1 randomly sampled view. We consider different **Res**olutions and model **Init**ialization strategies and compare to GS-LRM [92].

| Res. | Model | Init. | Alter. Canonical Views | | | Frame Interpolation | | | Two Rotating Cameras | | | Random Input Views | | |
|---|---|---|---|---|---|---|---|---|---|---|---|---|---|---|
| | | | PSNR | LPIPS | SSIM | PSNR | LPIPS | SSIM | PSNR | LPIPS | SSIM | PSNR | LPIPS | SSIM |
| 128 | 4D-LRM-Base | No | 27.374 | 0.068 | 0.937 | 27.287 | 0.068 | 0.937 | 25.496 | 0.075 | 0.924 | 24.174 | 0.112 | 0.893 |
| | 4D-LRM-Large | No | 28.489 | 0.055 | 0.949 | 28.440 | 0.055 | 0.949 | 26.312 | 0.064 | 0.934 | 25.023 | 0.096 | 0.905 |
| | 4D-LRM-Large | Yes | **29.251** | **0.042** | **0.959** | **29.169** | **0.043** | **0.958** | **28.676** | **0.043** | **0.957** | **27.586** | **0.064** | **0.939** |
| 256 | GS-LRM (Per Frame) | - | 18.796 | 0.164 | 0.854 | - | - | - | 19.729 | 0.143 | 0.911 | 18.300 | 0.176 | 0.845 |
| | GS-LRM (All in) | - | 19.388 | 0.138 | 0.888 | 19.412 | 0.139 | 0.888 | 19.428 | 0.138 | 0.888 | 19.379 | 0.138 | 0.888 |
| | 4D-LRM-Base | No | 25.806 | 0.085 | 0.928 | 25.711 | 0.085 | 0.929 | 23.974 | 0.085 | 0.924 | 22.409 | 0.125 | 0.889 |
| | 4D-LRM-Large | No | 26.658 | 0.066 | 0.942 | 26.580 | 0.066 | 0.942 | 25.524 | 0.067 | 0.937 | 24.461 | 0.095 | 0.913 |
| | 4D-LRM-Free | Yes | 28.838 | 0.050 | 0.958 | 28.790 | 0.051 | 0.958 | 26.998 | 0.056 | 0.949 | 25.267 | 0.082 | 0.923 |
| | 4D-LRM-Large | Yes | **30.094** | **0.041** | **0.967** | **30.028** | **0.042** | **0.967** | **27.810** | **0.049** | **0.957** | **26.694** | **0.072** | **0.939** |

Figure 6: Qualitative examples of 4D-LRM taking views with missing frames, including **Frame Interpolation** (only half timestamps seen, 24 input views) and **Random Views at Random Frames** (24 input views from random views and times).

- **Frame Interpolation.** Two canonical views are provided at even-numbered frames, while all odd-numbered frames are omitted from the input. This setup is designed to evaluate the model's interpolation ability across time;

- **Two Rotating Cameras.** Two virtual cameras rotate from front to back, one sweeping from the left and the other from the right. The left-rotating camera provides views at odd-numbered frames and the other for even-numbered frames, creating a complementary dual-view sequence;

- **Random Input Views.** At each frame, a single view is randomly selected from the available camera positions. This setting introduces high variability that requires robustness to unstructured input.

**Baselines.** To the best of our knowledge, 4D-LRM is the first large-scale 4D reconstruction model that supports inputs from unconstrained views and timestamps, and enables rendering at arbitrary novel view-time combinations. The closest baselines are recent multi-view diffusion models [73, 76] and feedforward models [39] that support camera pose and time control, though these methods are not publicly available. For setups from moving cameras, we compare 4D-LRM to GS-LRMs run in a per-frame fashion or directly with all views jointly. We also include Shape of Motion (SoM) [69], an optimization-based method that models videos as causal, one-way motion trajectories. It represents scene motion using compact SE(3) bases, enforcing consistent forward movement throughout dynamic scenes. Following its guidelines, we manually segment the dynamic region and run 3,000 optimization iterations. In addition, we perform systematic ablation studies on 4D-LRM, evaluating two model scales: **Large** (300M parameters; 1024 hidden size, 24 layers, 16 attention heads) and **Base** (85M parameters; 768 hidden size, 12 layers, 12 attention heads). We also explore the effects of different **Res**olution settings and **Init**ialization strategies, i.e., whether initializing the Transformer with weights from GS-LRM improves performance, using the same training setup and number of steps. Unless otherwise specified, we use 4D-LRM-Large with initialization by default. More design choices are discussed with training-time scaling in Section 4 and Appendix B.4.

Table 3: Comparison between 4D-LRM and GS-LRM under varying numbers of input views per frame (VPF). **Rand.**: randomly selected input views; **Canon.**: 4 canonical input views at each frame.

| Input | VPF | Model | Objaverse4D (Test) | | | Consistent4D (Re.) | | |
|---|---|---|---|---|---|---|---|---|
| | | | PSNR | LPIPS | SSIM | PSNR | LPIPS | SSIM |
| Rand. | 1 | GS-LRM | 18.738 | 0.165 | 0.852 | 17.201 | 0.166 | 0.846 |
| | | 4D-LRM | **28.343** | **0.040** | **0.964** | **30.513** | **0.036** | **0.972** |
| | 2 | GS-LRM | 22.252 | 0.105 | 0.898 | 22.425 | 0.097 | 0.904 |
| | | 4D-LRM | **28.622** | **0.051** | **0.957** | **30.601** | **0.035** | **0.973** |
| | 3 | GS-LRM | 24.381 | 0.083 | 0.917 | 24.661 | 0.079 | 0.924 |
| | | 4D-LRM | **28.212** | **0.052** | **0.954** | **30.554** | **0.035** | **0.972** |
| | 4 | GS-LRM | 26.118 | 0.070 | 0.933 | 26.126 | 0.070 | 0.935 |
| | | 4D-LRM | **27.940** | **0.053** | **0.953** | **30.445** | **0.034** | **0.972** |
| Canon. | 4 | GS-LRM | 28.710 | 0.047 | 0.962 | 30.067 | **0.038** | 0.967 |
| | | 4D-LRM | 27.839 | 0.055 | 0.952 | **30.850** | 0.039 | 0.968 |

Table 4: Application to 4D generation on the original Consistent4D benchmark. ‡Using ground truth multi-view reference in the first frame as the skyline.

| Model | PSNR | LPIPS | SSIM | FVD | CLIP |
|---|---|---|---|---|---|
| Consistent4D [28] | - | 0.160 | - | 1,133.44 | 0.87 |
| DG4D [52] | - | 0.160 | - | - | 0.87 |
| 4Diffusion [89] | - | 0.165 | - | - | 0.88 |
| Efficient4D [45] | - | 0.130 | - | - | 0.92 |
| GaussianFlow [22] | - | 0.140 | - | - | 0.91 |
| 4DGen [85] | - | 0.130 | - | - | 0.89 |
| STAG4D [88] | - | 0.130 | - | 992.21 | 0.91 |
| SV4D [78] | - | 0.129 | - | 677.68 | 0.93 |
| L4GM [53] | - | 0.120 | - | 691.87 | **0.94** |
| 4D-LRM-Large | 20.094 | 0.138 | 0.885 | 1,063.88 | 0.87 |
| 4D-LRM-Free | **23.777** | **0.117** | **0.916** | **677.58** | **0.94** |
| 4D-LRM-Free‡ | 26.118 | 0.055 | 0.947 | 674.59 | 0.96 |

Figure 7: Visual comparisons of 4D-LRM(-Free) to generation-based 4D models. For fair comparisons, we initialize each model with the first frame with the ground truth multi-view images.

**Metrics.** We follow previous work to adopt PSNR [5], SSIM [72], LPIPS [93] metrics. For a broad coverage of evaluation views, we evaluated and averaged the metrics on 4 canonical views and 1 randomly sampled view at each frame.

### 3.3 Main Results: 4D Reconstruction

On both the re-rendered Consistent4D dataset (Table 1) and the Objaverse4D dataset (Table 2), 4D-LRM demonstrates strong performance across a variety of camera configurations. Among the four tested setups, alternating canonical views provide the greatest view coverage over time and represent the least challenging configuration, under which 4D-LRM achieves PSNR scores exceeding 30. Notably, 4D-LRM remains robust even under more difficult settings, such as when half of the frames are omitted, or in configurations with limited spatiotemporal coverage, including two rotating cameras and random input views. We also find that increasing model size leads to improved performance under identical data and training steps. Additionally, initializing the Transformer with weights from GS-LRM further enhances performance and accelerates convergence.

The above experiments use 24 input views over a 24-frame motion sequence, corresponding to a only one view per frame. Since GS-LRM is designed for sparse-view reconstruction, we further compare it with 4D-LRM under denser input conditions. Specifically, we evaluate both models using: (1) multiple randomly selected input views per frame, and (2) four canonical views per frame, with the task of rendering a single randomly chosen novel view per frame. The results of these comparisons are presented in Table 3. We observe that GS-LRM performs comparably only when sufficient view coverage is available, as illustrated in Figure 5. In contrast, 4D-LRM consistently outperforms per-frame 3D reconstruction methods by leveraging spatial and temporal information jointly. As shown in Table 1, while SoM performs fair under structured settings such as canonical or rotating views, it struggles in the random input view scenario due to its limited capacity to handle unconstrained spatiotemporal inputs. This highlights the advantage of 4D-LRM's learned, generalizable representation for arbitrary view-time combinations. Finally, qualitative results in Figure 6 highlight 4D-LRM's ability to generalize to novel objects and interpolate effectively over time, even when input frames are missing.

### 3.4 Application: 4D Generation.

We demonstrate that 4D-LRM can be extended to a 4D generation setup. Specifically, we chain 4D-LRM (fine-tuned with free Gaussians) with SV3D [65], and compare it against existing generation-

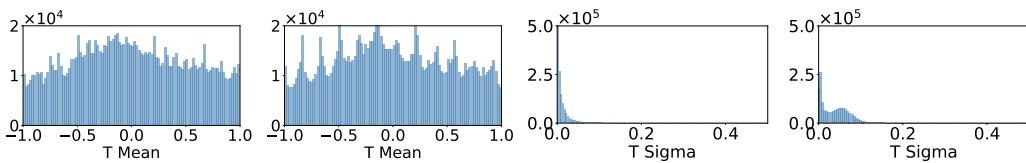

(a) $\mu_t$ w/o interpolation.    (b) $\mu_t$ w/ interpolation.    (c) $\Sigma_t$ w/o interpolation.    (d) $\Sigma_t$ w/ interpolation.

Figure 8: We visualize the distributions of $\mu_t$ and $\Sigma_t$ under **Alternating Canonical Views** and **Frame Interpolation** setups on 24 frames with the same dynamic object.

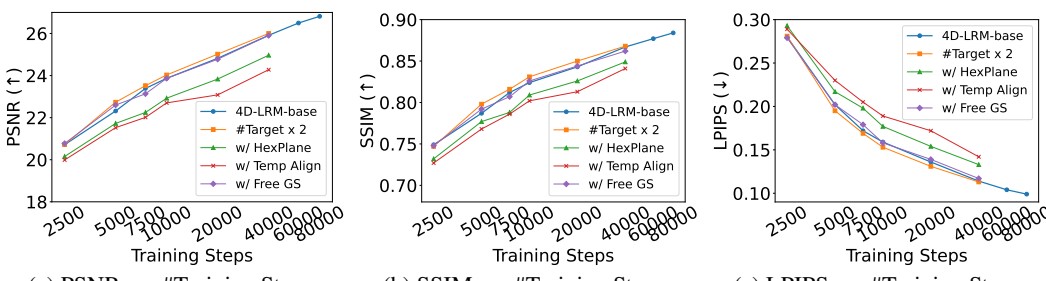

(a) PSNR vs. #Training Steps.    (b) SSIM vs. #Training Steps.    (c) LPIPS vs. #Training Steps.

Figure 9: Training-time scaling curves. Tested on Consistent4D (re-rendered). We compute the PSNR, SSIM and LPIPS for different number of different setups, with a 4D-LRM-base as the model.=

based methods. When paired with a diffusion model as a generative prior, 4D-LRM outperforms baseline 4D generation approaches on the original Consistent4D benchmark. This improvement is due to 4D-LRM's ability to produce much more faithful reconstructions of dynamic objects, even in the presence of motion ambiguity. Since diffusion models introduce high variance, we provide a comparison in Figure 7, where we evaluate 4D-LRM alongside other generative 4D models [53, 78] using ground-truth multi-view inputs from the first frame to ensure a fair comparison. Moreover, the core 4D-LRM model is efficient, requiring less than 1.5 seconds per forward pass; the primary computational bottleneck lies in the diffusion model.

## 4 Analyses and Discussions

**Why Can 4D-LRM Interpolate Frames?** 4D-LRM demonstrates strong frame interpolation capabilities, which aligns with its design: time is modeled as a continuous distribution rather than as discrete steps. To better understand this behavior, we analyze the 4DGS primitives predicted for the first 24 frames of the *Guppie* object in Consistent4D under two settings: Alternating Canonical Views (24 known timestamps) and Frame Interpolation (12 known timestamps). We visualize the distributions of the temporal mean $\mu_t = \mu_4$ and variance $\Sigma_t = \Sigma_{4,4}$ in Figure 10. Interestingly, when some timestamps are missing, 4D-LRM learns to reallocate certain Gaussians toward these missing regions, effectively filling the temporal gaps. Moreover, in the interpolation setting, the predicted 4DGS primitives tend to have larger $\Sigma_t$, increasing their temporal support. This allows each Gaussian to influence a broader range of neighboring timestamps after sampling, improving interpolation quality and temporal coverage.

**Training-Time Scaling.** To understand how different design considerations affect the training efficiency, we provide the scaling behavior with the following configurations to 4D-LRM-Base.

- **4D-LRM-Base**: Transformer with 768 hidden dimensions, 12 layers, and 12 attention heads, trained with 12 random input views and 12 random target views. No free Gaussians.
- **# Target $\times$ 2**: Trained with 12 random input views and 24 random target views.
- **w/ Hexplane**: Instead of unified space-time representation, Wu et al. [75] proposed an alternative 4DGS representation with decomposed neural voxel encoding inspired by HexPlane [3].
- **w/ Temp Align**: Similar to the idea of pixel-aligned Gaussians, we force $\mu_T$ to the input frame time, reducing the parameterization to $\dim_{4DGS} = 19$.
- **w/ Free GS**: Trained with $N = 1024$ free Gaussian tokens from scratch.

We observe that increasing the number of target views slightly improves convergence speed, though at the cost of increased iteration time. Introducing free Gaussians from scratch does not significantly impact reconstruction quality but substantially slows down training. Additionally, we find that the 4DGS representation from [75] is less expressive than the unified space-time formulation proposed

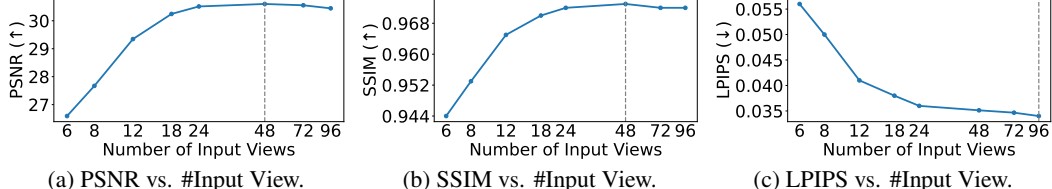

Figure 10: Inference-time scaling curves. Tested on Consistent4D (re-rendered). We compute the PSNR, SSIM, and LPIPS for different numbers of randomly selected input views.

by [82], which informed our final design choice. We also note that enforcing strict temporal alignment degrades performance, whereas pixel alignment improves reconstruction quality. This supports our earlier observation that 4D-LRM effectively redistributes Gaussians to unseen time intervals to handle sparse temporal supervision.

**Inference-Time Scaling.** Finally, we analyze inference-time scaling as the number of input views varies. In terms of PSNR and SSIM, performance improves with more input views and peaks at 48, after which it begins to decline slightly. We attribute this to two factors: (1) excessive Gaussians may overcrowd the 4D representation, reducing its quality, and (2) the Transformer struggles with very long input sequences. This observation suggests a promising future direction: designing 4D-LRM variants that can handle longer contexts with hybrid models [98] and incorporate test-time training [11, 94].

## 5 Related Work

Prior work on 4D modeling generally falls into three broad directions: optimization-based, geometry-based, and generation-based, each shaped by different assumptions, data requirements, and target applications. The first direction is optimization-based, mostly covered in previous discussions of 4D representations, where methods reconstruct dynamic scenes by optimizing per-object or per-scene representations using multi-view video [3, 28, 82, 69]. While capable of producing high-quality reconstructions, they are typically constrained by the need for dense spatial and temporal supervision. To improve generalization and reduce test-time cost, recent methods incorporate depth supervision or lightweight tuning [95, 62]. DyST [56] adopts a fully feed-forward approach, learning a latent decomposition of content, dynamics, and pose from monocular videos via a Transformer. However, it models space-time implicitly and remains limited in novel view synthesis quality. Although recent work has explored adapting LRMs for 4D asset generation [53] and scene-level reconstruction with limited input and target camera dynamics [81, 39], extending LRMs to general 4D reconstruction remains challenging, particularly when considering any target view at any time from sparse multi-views and missing timestamps. The second direction is geometry-based, which aims to estimate dynamic scene geometry, such as depth, flow, or camera motion, directly from input videos. Inspired by static sparse-view geometry methods like DUSt3R [71], recent work has extended this paradigm to dynamic scenes [90, 19, 70]. These methods often incorporate correspondence-based supervision or monocular depth priors to recover frame-wise geometry or trajectories. Unlike optimization-based approaches, they do not model the full spatiotemporal volume and are not intended for novel view or time synthesis. The third direction is generation-based, which leverages video generative models to synthesize perceptually plausible 4D assets. This includes both explicit 4D asset synthesis [53, 78, 83] and controllable video generation [36, 73, 76]. These methods reduce dependence on multi-view inputs by relying on learned priors, typically from large-scale video diffusion models. However, they are often computationally intensive at inference time, prompt-sensitive [36], and mostly limited to monocular inputs. Reconstructing faithful 4D geometry from a single-view video remains fundamentally ill-posed due to motion ambiguity [83]. Our goal is to learn a generic space-time representation that reconstructs an object from a few views at some time points, to any view at any time. We include an expanded related work section in Appendix due to the page limitation.

## 6 Conclusion

This work introduces 4D-LRM, the first large-scale 4D reconstruction model capable of processing unconstrained views and timestamps to render arbitrary novel view-time combinations. By learning a unified spatiotemporal representation and directly predicting per-pixel 4D Gaussian primitives from posed image tokens over time, 4D-LRM enables fast, high-quality rendering at, in principle, infinite frame rates.

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

# A    Related Work (Expanded)

**3D Representations.** At the heart of 3D reconstruction lies the choice of scene representation. Classical multi-view geometry methods [49, 57, 21, 1, 54] rely on calibrated images and epipolar constraints to recover structure via structure-from-motion (SfM; [49]) and multi-view stereo (MVS; [50, 21, 55]). While robust under densely sampled views, these pipelines degrade in the presence of occlusion, textureless surfaces, or wide baselines. To improve generalization, recent geometry-based models [71, 35] extend reconstruction to unposed and sparse view settings by leveraging large-scale pretraining. In parallel, neural optimization approaches fit implicit representations per scene, such as neural radiance fields (NeRF; [44, 4, 8, 15]) or signed distance functions [46, 42]. In this work, we adopt Gaussian Splatting (GS; [31, 25, 87]) as our 3D representation, which prioritizes efficiency and real-time performance through explicit point-based representations and offers faster rendering and competitive quality with minimal optimization per scene.

**4D Representations.** Modeling dynamic scenes in 4D has evolved from implicit, per-scene neural fields to more efficient, explicit spatio-temporal representations. Early methods [51, 47, 63] extend NeRF to dynamic scenes by conditioning on time or deformation fields, but they remain computationally heavy and lack real-time inference. A more practical shift arrives with HexPlane [3] and K-Planes [20], which decompose the 4D volume into planar factorizations. Concurrent with these NeRF-based approaches, dynamic Gaussian Splatting emerges as an explicit, high-performance framework for dynamic scenes [18, 33, 40]. D3GA [97] adapts this paradigm to human avatars by embedding Gaussian splats into deformable tetrahedral cages [84], allowing anatomical consistency and real-time control via joint-driven pose signals. Notably, 4D-GS [75] represents dynamic scenes using a canonical set of 3D Gaussians combined with 4D voxel-based features. These voxel encodings, inspired by the spatio-temporal factorization of HexPlane [3], are decoded by a lightweight MLP to predict per-Gaussian deformations across time. In contrast, we adapted 4DGS [82], which introduces a more elegant and unified formulation by directly parameterizing 4D Gaussians over space and time, allowing anisotropic deformation and smooth, time-varying appearance via 4D spherical harmonics.

**Feed-Forward Reconstruction Models.** Recent advances in generalizable radiance field-based methods have achieved state-of-the-art quality in novel view synthesis by leveraging NeRF-style volume rendering [86]. These approaches typically employ 3D-to-2D geometric projections to sample per-view image features, using architectural priors such as epipolar feature sampling [68, 59, 58] or plane-swept cost volumes [7, 30, 41] inspired by MVS. In contrast, we explore a simpler and more flexible design: a large Transformer-based model without explicit 3D inductive biases, which directly regresses Gaussian primitives. Parallel to radiance field approaches, a separate line of research investigates feed-forward, geometry-centric reconstruction models [61, 80], building upon DUSt3R [71] and leveraging large-scale training. This work aligns more closely with large reconstruction models (LRMs), which have recently emerged as a unified framework for producing view-consistent 3D reconstructions. These models are trained on massive 3D datasets and use triplane-based NeRFs [37, 24, 79, 67, 26] or 3D Gaussian Splatting [92, 60, 77, 98] to encode strong priors over shape and appearance, enabling high-quality reconstruction from just a few posed views. While early efforts have begun extending LRMs to generate 4D assets [53], we present the first LRM for general 4D reconstruction that can handle sparse multi-views and missing timestamps.

# B    4D-LRM Implementation and Training Details

## B.1    4DGS Parameterizing and Initialization

Given the decoded 4D Gaussian parameter $\mathbf{g} \in \mathbb{R}^{20}$, we split it into $(\mathbf{g}_{xyz} \in \mathbb{R}^3, \mathbf{g}_t \in \mathbb{R}, \mathbf{g}_{rgb} \in \mathbb{R}^3, \mathbf{g}_{scale,xyz} \in \mathbb{R}^3, \mathbf{g}_{scale,t} \in \mathbb{R}, \mathbf{g}_{rotation,left} \in \mathbb{R}^4, \mathbf{g}_{rotation,right} \in \mathbb{R}^4, \mathbf{g}_{opacity} \in \mathbb{R})$.

**Space and Distance.** Given the ray origin $\text{ray}_o$, direction $\text{ray}_d$, and a distance scalar $\delta$, the pixel-aligned Gaussian center in space along the ray is computed as $\mu_{xyz} = \text{ray}_o + \delta \cdot \text{ray}_d$. To derive $\delta$ from the decoded 4D Gaussian primitive $(\mathbf{g}_x, \mathbf{g}_y, \mathbf{g}_z)$, we adopt an interpolation scheme bounded by two empirically determined depth limits, $\delta_{near}$ and $\delta_{far}$, which define the permissible depth interval along each ray. This range constrains the model's predictions to lie within a spatially valid and semantically meaningful region of 3D space, consistent with the geometry captured during training.

$$\omega = \text{sigmoid}\left[(\mathbf{g}_x + \mathbf{g}_y + \mathbf{g}_z)/3\right], \tag{4}$$
$$\delta = (1 - \omega)\,\delta_{near} + \omega\,\delta_{far}. \tag{5}$$

Following the setup in [92], we set $\delta_{\text{near}} = 0.1$ and $\delta_{\text{far}} = 4.5$. The predicted $\mu_{\text{xyz}}$ values are further clipped to the range $[-1, 1]^3$.

**Scale and Opacity.** We follow the default activation functions used in 3DGS to ensure the predicted scale and opacity values fall within valid ranges: all scales are mapped to $\mathbb{R}^+$ and opacity to $(0, 1)$. Specifically, we apply the exponential activation to scale, mapping real-valued inputs to positive values, and the sigmoid activation to opacity. We observe similar training dynamics reported in [92], that the learned scale of 3D Gaussians can become excessively large. In such cases, the Gaussian degenerates into a highly anisotropic distribution, resembling a thin stick or line stretched across space and time. This can lead to unstable training dynamics, slow convergence, and temporal ghosting artifacts. To mitigate this, we apply constant biases to the Transformer's output to shift the initialization, and we clip the predicted scales to remain within a reasonable range.

$$\text{scale}_{xyz} = \min\{\exp\left(\mathbf{g}_{\text{scale,xyz}} - 2.3\right), 0.3\}, \tag{6}$$

$$\text{scale}_t = \min\{\exp\left(\mathbf{g}_{\text{scale,t}} - 2.3\right), 1.0\}, \tag{7}$$

$$\text{opacity} = \sigma(\mathbf{g}_{\text{opacity}} - 2.0), \tag{8}$$

All hyperparameters are empirically chosen and primarily serve to stabilize training. We observe that model performance is relatively insensitive to their specific values.

**Rotation.** We predict unnormalized quaternions and apply L2 normalization to ensure they lie on the unit hypersphere, producing valid unit quaternions for rotation. This approach simplifies optimization, as the model can freely output real-valued 4D vectors while normalization guarantees valid rotations. Following [82], we use a pair of $q_l = (a, b, c, d)$ and $q_r = (p, q, r, s)$ for the left and right unit quaternions, respectively. We use them to represent isotropic rotations in a symmetric form. $R$ can be constructed by:

$$R = L(q_l)R(q_r) = \begin{pmatrix} a & -b & -c & -d \\ b & a & -d & c \\ c & d & a & -b \\ d & -c & b & a \end{pmatrix} \begin{pmatrix} p & -q & -r & -s \\ q & p & s & -r \\ r & -s & p & q \\ s & r & -q & p \end{pmatrix}. \tag{9}$$

**Spherical Harmonics / RGB.** A 3D Gaussian stores a set of spherical harmonics (SH) coefficients to represent view-dependent color, along with a scalar opacity value $\alpha$. 4DGS extends this representation to enable both view-dependent appearance and its temporal evolution, by incorporating a time-variant extension of the SH basis. This allows the appearance of each Gaussian to change smoothly over both viewpoint and time. In our implementation, we directly interpret the model's output as the zero-order SH coefficients, following the convention used in 4DGS [82]. For simplicity, we do not include higher-order SH terms in this work.

## B.2 Differentiable Rasterization and Deferred Rendering

In the rendering process, given a pixel $(u, v)$ in an image $\mathbf{I}$ at time $t$, along with the camera's extrinsic matrix $E$ and intrinsic matrix $K$, the pixel color $\mathbf{I}(u, v, t)$ is computed by blending the contributions of all visible conditional 3D Gaussians. We build on the tile-based rasterization pipeline introduced in 3DGS and 4DGS [31, 82], and adopt deferred backpropagation [91] during rendering to reduce GPU memory consumption. We describe more details for completeness.

**Filtering.** At inference time, the final opacity of each conditional 3D Gaussian is weighted by its temporal marginal $p(t)$, which reflects its relevance at the rendered time step. To improve rendering efficiency and visual clarity, we apply two filtering strategies: (1) Gaussians with marginal probability $p(t) < 0.05$ are removed, and (2) Gaussians whose weighted opacity $\alpha < 0.05$ are removed. These filters are applied only during inference. Applying them during training would prematurely eliminate potentially useful Gaussians, leading to degraded convergence or dead Gaussians that fail to get optimized in training.

**Rasterization.** These filtered Gaussians are projected onto the image plane and sorted in front-to-back order based on their depth. For each pixel, the final color is computed using alpha blending, where the contribution of each Gaussian is weighted by its projected 2D density $p_i(u, v, t)$, its opacity $\alpha_i$, and its view-dependent color $c_i(d_i, t)$. Additionally, a transmittance term $\prod_{j=1}^{i-1} [1 - p_j(u, v, t)\alpha_j]$

**Algorithm 1** Image-to-4DGS Pseudo Code.

```
# Input dimensions:
# b = batch size; v = number of views; h, w = image height and width

# Input tensors:
# images : [b, v, h, w, 3] # RGB image frames
# frame_time : [b, v, 1] # Frame timestamp per view
# extrinsics : [b, v, 4, 4] # Camera-to-world (c2w) transformation matrices
# intrinsics : [b, v, 4] # Camera intrinsics (fx, fy, cx, cy)

# Output tensors (4DGS parameters):
# xyzt : [b, *, 4] # 4D Gaussian centers (x, y, z, t)
# rgb : [b, *, 3] # RGB color
# scale : [b, *, 4] # Anisotropic Gaussian scale (3D space + time)
# rotation : [b, *, 4, 4] # Rotation matrix
# opacity : [b, *, 1] # Opacity value

# Augment and patchify input for 4D representation
x_grid, y_grid = meshgrid(h, w) # [h, w]
ray_dir_cam = compute_camera_rays(x_grid, y_grid, intrinsics) # [b, v, 3, h, w]
ray_dir_world = transform_directions(ray_dir_cam, extrinsics) # [b, v, 3, h, w]
ray_origin = extract_camera_origin(extrinsics) # [b, v, 3, h, w]
o_dot_d = dot_product(-ray_origin, ray_dir_world, dim=2) # [b, v, 1, h, w]
nearest_pts = ray_origin + o_dot_d * ray_dir_world # [b, v, 3, h, w]

# Concatenate and patchify augmented images into transformer input
x = concatenate(
    normalize_rgb(images), # [b, v, 3, h, w], RGB scaled to [-1, 1]
    normalize_t(frame_time), # [b, v, 1, h, w], time scaled to [-1, 1]
    ray_dir_world, # [b, v, 3, h, w]
    nearest_pts # [b, v, 3, h, w]
) # Final: [b, v, 10, h, w]
x = patchify(x, patch_size=8) # [b * v, num_patches, patch_dim]

# Transformer
x = linear(x) # [b, v * num_patches, hidden_dim]
x = transformer(LN(x)) # LayerNorm + Transformer
x = depatchify(LN(x), out_dim=20) # [b, v * h * w, 20]

# 4DGS Parameterization
# Step 1: Split the transformer output into individual 4DGS fields
xyz, t, rgb, scale_xyz, scale_t, rotation_left, rotation_right, opacity = \
    split(x, sizes=[3, 1, 3, 3, 1, 4, 4, 1], dim=-1)

# Step 2: Compute center position (xyz + t)
w = sigmoid(norm(xyz)) # Soft depth interpolation weight
delta = near * (1 - w) + far * w # Range interpolation [near, far]
xyz = ray_origin + ray_dir_world * delta # 3D center point
xyzt = concatenate(xyz, t) # [b, v * h * w, 4]

# Step 3: Compute scale (clipped exp)
scale_xyz = clip(exp(scale_xyz - 2.3), max=0.3) # Spatial scale
scale_t = clip(exp(scale_t - 2.3), max=1.0) # Temporal scale
scale = concatenate(scale_xyz, scale_t) # [b, v * h * w, 4]

# Step 4: Normalize quaternions
q_l = normalize(rotation_left) # [b, v * h * w, 4]
q_r = normalize(rotation_right) # [b, v * h * w, 4]

# Step 5: Construct rotation matrix R = L(q_l) * R(q_r)
L = build_left_quaternion_matrix(q_l) # [b, v * h * w, 4, 4]
R = build_right_quaternion_matrix(q_r) # [b, v * h * w, 4, 4]
rotation = matmul(L, R) # [b, v * h * w, 4, 4]

# Step 6: Compute opacity
opacity = sigmoid(opacity - 2.0) # [b, v * h * w, 1]

# Final 4DGS outputs
return xyzt, rgb, scale, rotation, opacity
```

models the amount of light that reaches the $i$-th Gaussian after being attenuated by all previous ones. This formulation enables differentiable, order-dependent compositing. Yang et al. [82] noted that $p_i(u, v, t)$ can be factorized as the product of a conditional and a marginal probability at time $t$:

$$\mathcal{I}(u, v, t) = \sum_{i=1}^{N} p_i(t) p_i(u, v|t) \alpha_i c_i(d, t) \prod_{j=1}^{i-1} [1 - p_j(t) p_j(u, v|t) \alpha_j]. \tag{10}$$

To compute the image-space density $p_i(u, v|t)$, we start with the 4-channel space-time features in the order of xyzt, and compute the conditional 3DGS:

$$\mu_{i,xyz|t} = \mu_{i,1:3} + \Sigma_{i,1:3,4}\Sigma_{i,4,4}^{-1}(t - \mu_{i,4}),$$
$$\Sigma_{i,xyz|t} = \Sigma_{i,1:3,1:3} - \Sigma_{i,1:3,4}\Sigma_{i,4,4}^{-1}\Sigma_{i,4,1:3} \tag{11}$$

We approximate the projection of a 3D Gaussian $\mathcal{N}(\mu_i, \Sigma_i)$ using a linearized perspective transformation as is in [31]. The resulting 2D Gaussian is

$$p_i(u, v) \sim \mathcal{N}(\mu_{i,uv}, \Sigma_{i,uv}), \tag{12}$$

where the mean and covariance are computed as $\mu_{i,uv} = \mathrm{Proj}(\mu_{i,xyz|t}, E, K)_{1:2}$ and $\Sigma_{i,uv} = (JE\Sigma_{i,xyz|t}E^\top J^\top)_{1:2,1:2}$, with $\mathrm{Proj}(\cdot, \cdot, \cdot)$ denoting projection from world to image coordinates using extrinsic $E$ and intrinsic $K$, and $J$ the Jacobian of the perspective projection at $\mu_{i,xyz|t}$.

## B.3 Dataset Curation

To enable large-scale training, we construct a 4D dataset derived from Objaverse [13], which provides a subset of animated 3D assets.[1] However, the raw dataset is not directly suitable for 4D modeling: object motions are often inconsistent, and the dataset contains duplicates and artifacts. We build upon the filtered subset curated by Diffusion4D [38], which removes static objects and unstable motion sequences. For each object, we render a 24-frame video from diverse camera trajectories. If the original animation has fewer frames, we pad by repeating the last frame. The views include four canonical static views (front, back, left, right), elevated moving trajectories, and randomized orbits at varying distances, similar to [92]. This design encourages robustness to both viewpoint and motion variation. To further ensure quality, we compute the maximum L1 distance across frames for each sequence to select a high-quality subset with sufficient but not overly aggressive motion. Our final dataset contains 3,000 high-quality animated objects (HQ4D) selected from 32,000 animated objects (4D). We augment the dataset with 783,000 static 3D objects from Objaverse by treating each as a 24-frame video, applying minor frame-by-frame displacements along a single random direction. During pretraining, we sample from HQ4D, 4D, and 3D with a mixing ratio of 200:50:1.

## B.4 Training Details

We keep most settings identical to GS-LRM [92] so we can initialize 4D-LRM training upon it. Below, we describe the details for completeness. We use a patch size of $8 \times 8$ for the image tokenizer. **4D-LRM-Large** employs a 24-layer Transformer with a hidden dimension of 1024, 16 attention heads, and a two-layer MLP with GeLU activation. **4D-LRM-Base** uses 12 layers with a hidden dimension of 768 and 12 attention heads, sharing the same MLP design. All Transformer blocks are equipped with Pre-Layer Normalization and residual connections. Additionally, Layer Normalization is applied after the patchifying linear layer and before the unpatchifying linear layer to stabilize training. To enable efficient training and inference, we adopt Flash-Attention v2 [12] via the `xFormers` library [34], along with gradient checkpointing [10] and mixed-precision training using the BF16 data type [43].

# C Additional Results

## C.1 Evaluation on 3D Reconstruction

Table 5 reports performance on the GSO dataset [17], comparing various models under two resolutions. Notably, when adapting 4D-LRM for static 3D reconstruction by setting all timestamps to zero, we observe a modest drop in performance relative to GS-LRM at 256×256 resolution. Despite this, 4D-LRM still outperforms the LGM and many models evaluated at higher resolution. This suggests that the spatiotemporal representations learned by 4D-LRM remain effective for conventional 3D tasks, highlighting its versatility and robustness.

## C.2 Failure Cases

---

[1]We exclude Objaverse-XL [14] due to license restrictions.

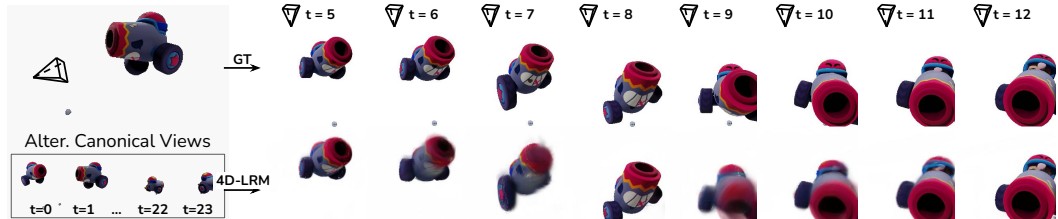

Figure 11: A typical failure case as 4D-LRM sometimes struggles with non-linear motion trajectories. When an object follows a non-linear path, the structures cannot be efficiently captured by a single ellipsoidal Gaussian. As a result, the model requires multiple Gaussians placed along the trajectory to approximate the motion, increasing complexity and often leading to artifacts if not properly aligned.

We provide a typical failure case in Figure 11. 4D-LRM still struggles with challenging 4D reconstruction scenarios involving self-occlusion and fast motion, which often result in temporal ghosting artifacts. These failures are primarily due to the limitations of both (1) under-training (the current model has not yet reached the saturation for scaling) as well as (2) the current Gaussian representation, which models appearance and motion as smooth, continuous functions. In the presence of rapid or discontinuous changes, such as objects moving in and out of occlusion or undergoing abrupt non-rigid deformation, the model

Table 5: Performance on the GSO dataset [17] for 3D reconstruction. 4D-LRM is evaluated by collapsing frame times to 0.

| Res. | Model | PSNR | LPIPS | SSIM |
|---|---|---|---|---|
| 512 | SparseNeus [41] | 20.62 | 0.199 | 0.836 |
| | Triplane-LRM [37] | 26.54 | 0.064 | 0.893 |
| | Mesh-LRM [74] | 27.93 | 0.081 | 0.925 |
| | GS-LRM [92] | 30.52 | 0.050 | 0.952 |
| 256 | LGM [60] | 21.44 | 0.122 | 0.832 |
| | GS-LRM [92] | 29.59 | 0.051 | 0.944 |
| | 4D-LRM | 27.35 | 0.061 | 0.929 |

cannot accurately localize or update the corresponding Gaussians in time. As a result, outdated Gaussians persist across frames, leading to visually noticeable residuals and motion trails. 4D-LRM sometimes falls short in non-linear trajectories. The kernel density of a Gaussian is ellipsoidal, it defines mass aligned with principal directions. When an object follows a non-linear path, the optimal support is curved or branched, not ellipsoidal. To approximate a non-linear trajectory, the model needs more Gaussians along the curve.

## C.3 Additional Qualitative Examples

We provide additional qualitative examples in Figure 12 and 13.

## D Limitations

We highlight the following future directions:

**Long Context.** Issues such as limited resolution, short video duration, and occlusion are fundamentally challenges of memory and long-range dependencies in sequence modeling. Although 4D-LRM achieves high-quality reconstruction from sparse posed images, several limitations remain. First, it cannot yet efficiently process hundreds of input images in a single forward pass. Second, its maximum training resolution is $256 \times 256$, though it generalizes up to $512 \times 512$. Unlike GS-LRM, which was fine-tuned at 512 resolution, fine-tuning 4D-LRM at this scale is significantly more expensive, requiring approximately 75 seconds per training step. A promising direction for future work is to develop high-resolution 4D reconstruction models capable of handling hundreds of 1K or 2K resolution inputs. This will require fundamental architectural advances, such as hybrid models for long-context handling [98] and test-time training strategies [11, 94].

**Removing 3D Inductive Bias.** Currently, 4D-LRM relies on posed images and explicitly learns 4D Gaussian primitives for rendering. To scale up 4D representations from in-the-wild videos, future work should aim to remove strong 3D inductive biases. This includes learning to reconstruct from unposed images [67, 27], and designing architectures that forgo explicit 3D representations such as NeRF or 3DGS [29, 27, 94].

**From Objects to Scenes.** Currently, 4D-LRM is not trained at the scene level, as the concept of "any view" is less well-defined, e.g., we cannot observe what lies behind walls. Although GS-LRM has shown that this architecture can scale to scene-level reconstruction, we currently lack

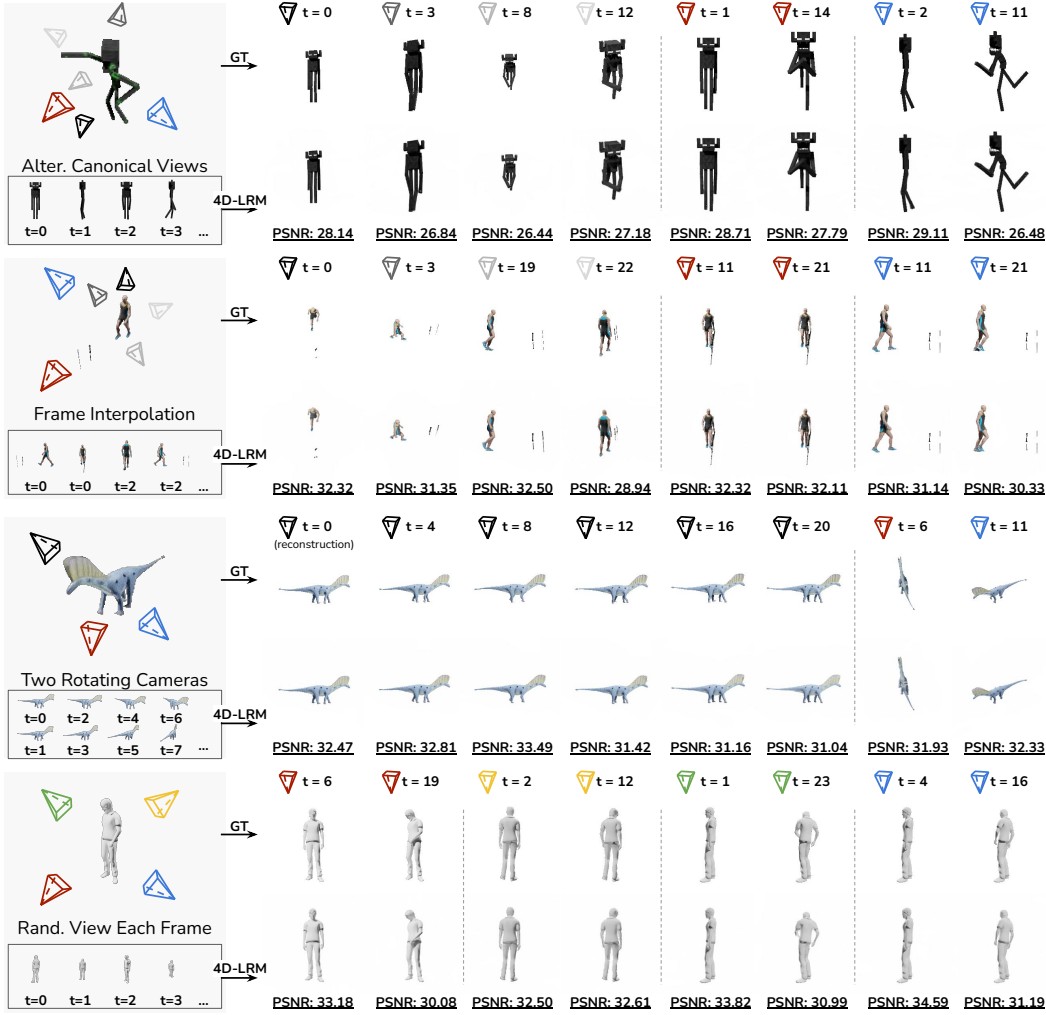

Figure 12: Qualitative examples of 4D-LRM under varying camera setups. We show the performance of 4D-LRM when taking input views captured with different camera configurations, demonstrating its robustness to diverse spatial arrangements and viewpoints.

a license-compliant, high-quality 4D scene dataset for training. Moreover, the data augmentation strategies used for object-level data do not directly transfer to scene-level setups. While we start to see attempts on this line with limited camera movement or domain-specific applications and limited input/target camera dynamics [81, 39], more future work should investigate both scalable 4D datasets and training methods for extending 4D-LRM to scene-level reconstruction.

# E    Broader Impact

While this paper does not explicitly address societal impact, the proposed 4D representation learning method has the potential to benefit a range of downstream applications, including robotics, AR/VR, and digital content creation, by enabling more accurate and efficient modeling of dynamic scenes. However, the work primarily focuses on technical contributions, and we do not identify immediate ethical concerns or direct social implications. As with any foundation model capable of detailed spatial-temporal understanding, future applications should consider issues of privacy, surveillance, and potential misuse, especially if deployed in real-world environments involving human data.

# F    Acknowledgment.

The authors would like to thank Ang Cao, Junyi Zhang, and Wenhao Chai for their helpful discussions and feedback.

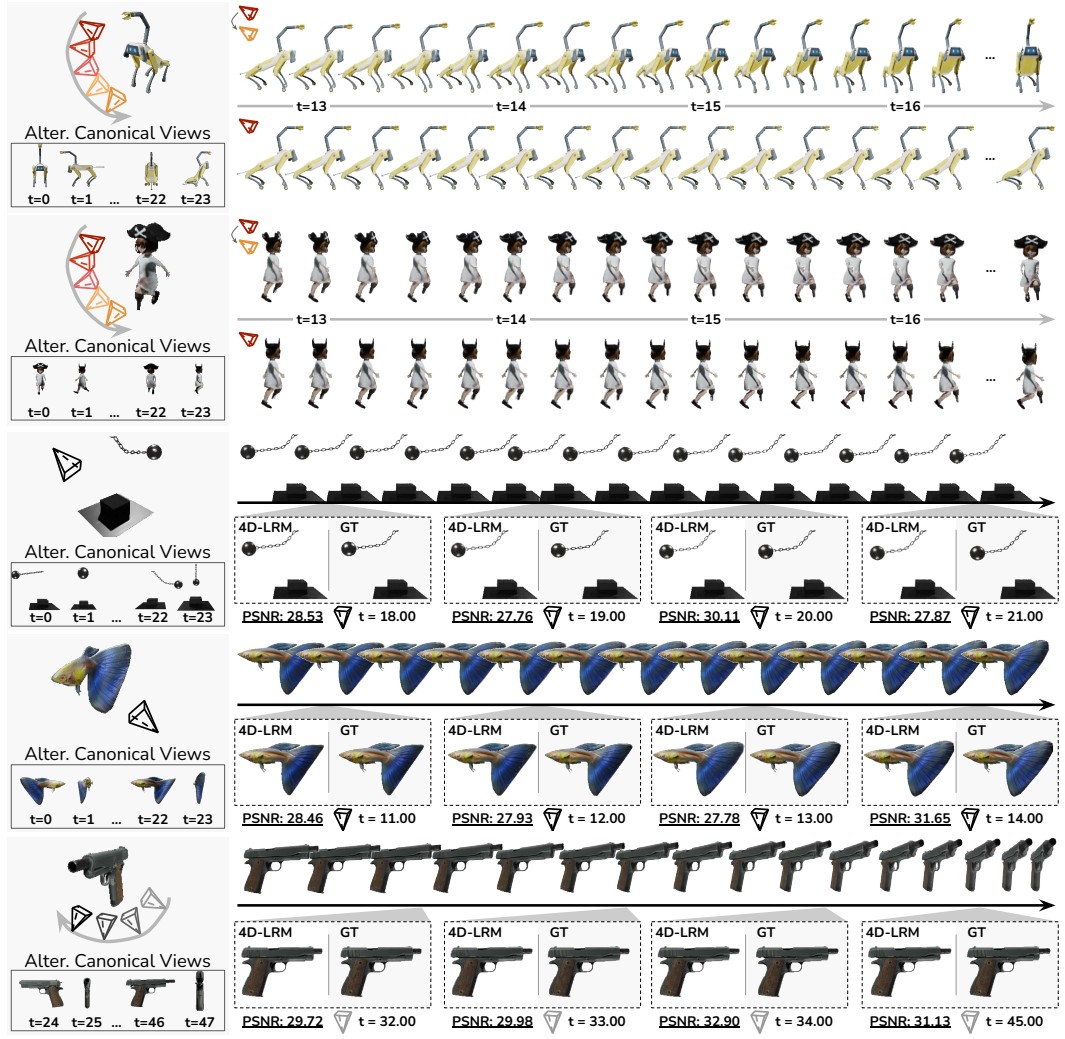

Figure 13: Additional frame interpolation examples. We insert $4\times$ denser frames between Alternating Canonical Views as input.

