# OpenReview forum: "4D-LRM: Large Space-Time Reconstruction Model From and To Any View at Any Time"
_NeurIPS.cc/2025/Conference — NeurIPS 2025 poster_

### Official Review · Reviewer_DFse · 2025-07-01

**Clarity:** 3
**Significance:** 2
**Originality:** 3
**Rating:** 4
**Confidence:** 4

**Summary:**

The paper proposes a transformer-based model to reconstruct 4DGS from view-varied and time-varied input images. By enforcing unified modeling of space and time inherited from 4DGS, the proposed method can theoretically render arbitrary novel viewpoints and timestep given unconstrained input views and timesteps. Extensive experiments are done on testset from Objaverse and Consistent4D to validate the superior reconstruction performance of the proposed method. However, similar to many previous work trained on Objaverse, the proposed method is still limited to the foreground of object-level scenes.

**Questions:**

-	It is pretty interesting that the proposed method uses a unified representation of space and time on the output side (4DGS), however, it uses a modeling as temporal + image patches on the input side. I’m curious that will this “asymmetric” modelling brings pixel-aligned Gaussian features? It would be great to have a visualization of learned per-pixel feature for Gaussian primitives on the spatial-temporal 4D grids.
-	In case sufficient views and timesteps are provided, how does the performance curves of optimization-based 4D reconstruction method look like? It would be great to have a similar observation like Figure 9 to see the upper bound.
-	Although the generative capability is supported through free Gaussians, will it have sampling diversity just like native generative models?

**Ethical Concerns:**

["NO or VERY MINOR ethics concerns only"]

**Final Justification:**

All my concerns have been addressed. I will keep my original rating towards acceptance

**Limitations:**

Yes

**Quality:**

3

**Strengths And Weaknesses:**

Strengths:
- The paper is well-written and organized with a clear illustration of key concept and difference to prior work (e.g., Figure 2)
- As the very few papers on generic 4D reconstruction (M-in-N-out), the proposed method finds a principled but scalable solution with limited assumptions on the input.
- Personally, I really like the design of introducing generative capacity by a set of optional learnable free Gaussian tokens. It works surprisingly well as shown in Table 4, especially in FVD.
- The performance of the proposed method is extensively evaluated by quantitative results under different test settings and model specifics.

Weaknesses:
- As a 4D reconstruction method, qualitative results (videos) of comparisons and ablation studies are expected in the supplementary, which brings more direct evaluation of the reconstruction quality especially temporal smoothness.
- The generalization to more general scenes is not validated. All the results are shown as object-level scenes. It somehow limits the usecases of the proposed method and makes it a bit unfair when compared with methods based on video diffusion models. The latter one can solve general 4D generation while the proposed method only focus on object-level scenes.

---

> ### Author Rebuttal · Authors · 2025-07-31
>
> We sincerely thank the reviewer for the encouraging and detailed feedback. We appreciate the recognition that the paper is "well-written and organized," with a "clear illustration of key concept and difference to prior work (e.g., Figure 2)." We are glad the reviewer found our approach to be a "principled but scalable solution with limited assumptions on the input," especially as one of the very few works tackling generic 4D reconstruction in an M-in-N-out setting.
>
> ## Weakness 1: Qualitative Results (Videos)
> We confirm that qualitative videos have already been included in the supplementary ZIP file. These provide visual comparisons of reconstruction quality and temporal smoothness. We did not include sample videos for every ablation study, as training all ablated variants to full convergence is computationally infeasible. However, we present a comprehensive set of quantitative ablations in Section 4.2, covering key design choices such as 4D representation, time encoding, initialization strategy, and number of target views. We also include training-time scaling curves to reflect convergence behavior and performance trends across variants.
>
> ## Weakness 2: Generalization
>
> We appreciate the suggestion and agree that real-world testing is important. To address this, we adapted the Plenoptic Video dataset, which contains 6 human motions in real indoor scenes. We preprocess them via center cropping and downsampling to 256×256 resolution with FOV 50 to match our training settings. For each sequence, we sample 24 frames across time and render to 6 novel views. We evaluate under two input settings: using raw frames vs. using SAM2 to segment the dynamic human.
>
> | Input Type | PSNR  | LPIPS | SSIM |
> |------------|-------|-------|------|
> | Raw        | 24.73 | 0.06  | 0.95 |
> | Segment    | 27.53 | 0.03  | 0.98 |
>
> While there is a performance drop compared to pure object-centric data (due to domain shift and background clutter), the results are non-trivial and demonstrate generalization, especially considering the small number of input views (24) without optimization.
>
> We note that Objaverse already includes real scene-normalized data, and prior work (e.g., GSLRM) has shown that object-level training can generalize to simple scene-level layouts. While we have not trained a full scene-level 4D-LRM due to the lack of license-permissive 4D scene datasets, we view this as promising future work. The underlying design of 4D-LRM is extensible to more complex real-world scenarios.
>
> ## Question 1: Gaussian Features
>
> ### On pixel-aligned Gaussian features:
> Yes, the predicted Gaussians are still pixel-aligned. During tokenization (Section 2.2), each pixel is augmented with Plücker coordinates (encoding camera pose) and timestamp, along with RGB. The spatial center of the Gaussian is derived from the corresponding ray direction and distance, see Appendix A.1 for details. While spatial alignment is preserved, temporal alignment is not enforced. As shown in our ablation (Section 4.2), forcing Gaussians to match input timestamps degrades performance.
>
> ### On feature visualization:
> Figure 7 already provides a case study illustrating temporal reallocation behavior: when timestamps are missing, 4D-LRM learns to shift Gaussians toward these gaps and increases their temporal support. This behavior supports our design choice to preserve pixel-level spatial alignment while allowing flexible temporal modeling.
>
> ## Question 2: Optimization-based 4D Reconstruction
>
> We agree that optimization-based 4D reconstruction methods can serve as useful upper bounds or baselines. We provide two supporting experiments with 256 resolution on Consistent4D:
>
> Shape of Motion (SoM) is included in our main experiment as an optimization-based baseline. SoM models 4D videos as causal, one-way trajectories, but it does not support frame insertion or multiple frames per timestamp without significant modification. We manually segment the dynamic region and run 3,000 optimization steps per object.
>
> | Model          | Alter. Canonical Views (PSNR / LPIPS / SSIM) | Two Rotating Cameras (PSNR / LPIPS / SSIM) | Random Input Views (PSNR / LPIPS / SSIM) |
> |----------------|----------------------------------------------|---------------------------------------------|------------------------------------------|
> | Shape-of-Motion | 25.586 / 0.055 / 0.906                       | 22.756 / 0.089 / 0.941                      | 17.637 / 0.208 / 0.875                   |
> | 4DLRM          | 32.177 / 0.028 / 0.980                       | 27.664 / 0.050 / 0.957                      | 27.990 / 0.057 / 0.954                   |
>
> For upper-bound analysis similar to Figure 9, we perform direct optimization of 4D Gaussians (4DGS) from the full set of input frames. However, this setup becomes unstable when the number of views is low (especially when <48), with high sensitivity to initialization and hyperparameters. We report the best results achieved for each input size, but emphasize that this is not the intended setting and use case of 4DGS and serves only as an empirical reference.
>
> | Model | #Views | PSNR  | LPIPS | SSIM |
> |-------|--------|-------|-------|------|
> | 4DLRM | 24     | 30.51 | 0.04  | 0.97 |
> | 4DLRM | 48     | 30.60 | 0.04  | 0.97 |
> | 4DLRM | 96     | 30.45 | 0.03  | 0.97 |
> | 4DGS  | 48     | 25.02 | 0.08  | 0.92 |
> | 4DGS  | 96     | 30.64 | 0.03  | 0.98 |
>
> ## Question 3: Sample Diversity
>
> Unfortunately no. As is also mentioned in the original LRM paper, Section 4.3.2, the model "is deterministic and is likely producing averaged modes of the unseens.” While free Gaussians increase flexibility and generative capacity, they do not introduce sampling diversity by design. To enable diversity, a promising future direction is to integrate 4D-LRM into a probabilistic or diffusion-based pipeline (such as Turbo3D), where the deterministic backbone can be paired with latent sampling mechanisms. We consider this an exciting avenue for extending our framework. We will mark this in the limitations section in the next version.

---

> > ### Comment · Reviewer_DFse · 2025-08-04
> >
> > Thanks for the detailed response from the authors. All of my concerns have been properly addressed. I will keep my original rating.

---

> > > ### Author Response · Authors · 2025-08-04
> > >
> > > Thank you for your thoughtful review and for confirming that all your concerns have been addressed!

---

### Official Review · Reviewer_eKZZ · 2025-07-02

**Clarity:** 2
**Significance:** 2
**Originality:** 2
**Rating:** 2
**Confidence:** 5

**Summary:**

This paper studies how to reconstruct dynamic 3D objects from posed, discrete space-time RGB frames. The input setting is object-centric RGB images captured at arbitrary times and poses (in contrast to continuous views or monocular videos). The output is a 4D Gaussian representation (a true 4D distribution, not a deformable one).

To generate this set of 4D distributions, multiple input images are first encoded with RGB, pose Plücker coordinates, and a time ID, and then fed into a simple transformer. Both pixel-aligned Gaussians are queried using the input patches, and additional free queries are introduced to fill in unobserved or missing areas. This simple transformer is trained via regression on filtered Objaverse sequences. Some visual results and ablation studies are presented, primarily using photometric reconstruction metrics.

**Questions:**

Why is this paper novel? Why is this task hard/non-trivial given the prior art?

Is the motion quality good?

Is this a realistic problem setup?

Is this research direction correct?

**Ethical Concerns:**

["NO or VERY MINOR ethics concerns only"]

**Final Justification:**

See my comment below

**Limitations:**

There is a section describing the limitations, including from object to scenes and resolutions, but compared to the above-mentioned questions, I believe these are minor points.

**Paper Formatting Concerns:**

Mostly okay, the only awkward point is that such an applied/engineering paper puts its related work at the end, which is not very common in the vision community.

**Quality:**

2

**Strengths And Weaknesses:**

**Strengths:**

- At a high level, 4D is a fundamental direction for computer vision in the coming years.
- The design of the network architecture is straightforward but reasonable.
- Despite the very limited data source (Objaverse 4D data is indeed simple and constrained), some “scaling” behavior is observed in the experiments, at least indicating the network’s fitting dynamics for this specific transformer.

---

**Weaknesses:**

- **Unrealistic problem setup and unclear novelty**: The paper emphasizes that reconstruction can happen at arbitrary poses and times. However, such settings are not reflective of real-world scenarios. In contrast, studying monocular video input and augmenting it to multiview is arguably a more “real” and practical problem. Therefore, the reviewer believes the premise of this paper may be fundamentally flawed. Moreover, there is no evidence that the model works on real data — how would one even obtain object-centric, multiview, discretely posed frames at different time steps in the real world? Using multiview camera setups? Given these concerns, the reviewer notes that prior work has explored reconstructing 4D deformable objects from monocular videos. Simply permuting input pairs and stacking them with a basic GS transformer does not constitute sufficient novelty.
- **Possibly wrong research direction**: Although the paper benefits from industrial-level resources, the authors themselves may have noticed the poor quality of the data. The relative motion in the sequences is minimal, making this model closer to a 3D LRM rather than a true 4D model. This limitation is not overcome by “scaling.” The bottleneck in this area is not in network design or output representation, but in the data itself. Fitting Objaverse is straightforward because its motion distribution is small and simple. This is unlike 3D generation/reconstruction, where challenges lie in the high variance of detailed geometry and texture. For 4D objects, motion is the core difficulty; otherwise, the task reduces to standard 3D reconstruction. Thus, building a regression model on Objaverse for 4D appears trivial. The real research direction should focus on how to scale such models using self-supervision or rendering-based supervision on real data without known geometry-appearance.
- **Conceptual flaws**: A key aspect of 4D modeling is leveraging temporal and camera smoothness — not discrete multiview observations at random poses. In contrast, monocular video is a more realistic and meaningful setup, rather than the “toy” setting used in this paper.
- **Generating Gaussian Splatting is not fundamental**: Stepping back, if we consider this as a simple supervised regression problem on Objaverse, it’s unclear why GS is necessary as a representation. Hopefully, the choice isn't just due to its popularity. GS is suitable for rendering, but if ground-truth 4D geometry and appearance are available, why not supervise directly in 3D? Why is rendering needed at all if the goal is not to bridge toward real-world data?
- **Lack of correspondence**: The representation uses  the real 4D Gaussians, not deformation-based dynamic Gaussians, so it fails to establish temporal correspondence — a critical, as important aspect of 4D reconstruction/generation alongside visual quality.
- **Low output quality (especially under 160 A100s)**: The motion is relatively simple and small, largely constrained by the limited distribution of the Objaverse dataset.
- **Incomplete literature review**: Many relevant works on 4D reconstruction and generation are missing; the paper lacks a comprehensive discussion of related literature.

---

> ### Author Rebuttal · Authors · 2025-07-31
>
> We thank the reviewer for the thoughtful feedback and for recognizing the high-level motivation of our work. We appreciate the acknowledgement that “4D is a fundamental direction for computer vision in the coming years,” and that our network architecture is “straightforward but reasonable.” We are also encouraged by the observation that scaling behavior was observed, suggesting the model’s fitting dynamics are promising for this transformer design. We address the reviewer’s comments and clarifications in detail below.
>
> ## Weakness 1/3 and Question 1/3: Problem Setup / Novelty / Conceptual Flaws
>
> ### Problem Setup / Conceptual Flaws
> Reconstruction at arbitrary poses and times is not only our training objective, but it is also the most general form of the 4D reconstruction problem. We agree that monocular video (and augmenting it to multiview) is a (but not the only) practical and well-motivated setting, as argued in prior work like L4GM. However, **it is only a special case of our any-to-any formulation**, which **we include as a part of our evaluation** but go beyond. Therefore, our setting is not a “toy setting” but the superset of monocular input.
>
> ### Real-World Applicability
> Our setup reflects a broader range of practical scenarios (see Section 3.3 and Figure 10 in the Appendix) studied in the vision community:
> - A few fixed cameras (e.g., surveillance cameras or motion capture),
> - A few continuous moving cameras (e.g., sports broadcasting, mobile robotics),
> - Single fixed monocular video augmented with multiview diffusion prior (e.g., 4D asset generation),
> - Sparse asynchronous recordings (e.g., crowd-sourced phone videos or event capture).
>
> These settings all involve varying degrees of spatial and temporal sparsity, which our model is explicitly designed to handle.
>
> ### Novelty
> We strongly disagree with the characterization that our method merely permutes input pairs and stacks them in a basic GS transformer. A core novelty of 4D-LRM is the unified space-time treatment using anisotropic 4D Gaussian primitives, enabling continuous-time interpolation, which is a capability not supported by prior feedforward 4D reconstruction methods, and achieved here in a simple, elegant design.
>
> ## Weakness 2/6 and Question 2/4: Research Direction / Motion
>
> ### Modeling vs. Data
> We disagree that the bottleneck lies solely in data and not in modeling. While high-quality dynamic 4D data is limited and remains a challenge, very few works have offered a clean and scalable solution for 4D reconstruction beyond generation tasks. Our work not only provides such a general solution, but also presents ablation studies analyzing which 4D representations are expressive and which model designs impact scaling behavior.
>
> ### Data Quality
> Although Objaverse is not ideal, it is the only available large-scale dataset with open licenses, and all existing baselines (including L4GM and SV4D) are trained on it. Within this shared training data for fair comparison, we show that model design and representation choices still matter: 4D-LRM outperforms baselines and supports more flexible input configurations.
>
> ### Motion
> Fundamentally, what matters is not the magnitude of relative motion, but the frame rate at which the model receives input. We agree that motion is a key challenge in 4D modeling, and this is why, in evaluation, we manually hold out 56 challenging objects with more complex motion as the test set. Our experiments also show against the reviewer’s hypothesis that “the task reduces to standard 3D reconstruction” by showing 4D-LRM outperforms strong per-frame 3D models by effectively leveraging spatiotemporal correlations across frames. This highlights that our model learns and exploits non-trivial spatiotemporal dynamics.
>
> ## Weakness 4: Generating Gaussian Splatting is not fundamental
> Our goal is to identify the most effective 4D representation for scaling large reconstruction models. We initially explored HexPlane-style representations but found them lacking. We adopted 4D Gaussian Splatting based on strong preliminary results, and we compared two different 4DGS parameterizations in Section 4.2, which informed our final design choice. At the time of submission, alternative approaches (e.g., diffusion-based or decoder-only models) had not been open-sourced or made reproducible, so GS was the most practical and validated representation for this setting.
>
> Regarding direct supervision in 3D, we respectfully discussed (line 26) recent feedforward methods that rely on monocular depth priors or explicit correspondences to recover per-frame geometry or trajectories. These methods are not designed for novel view synthesis, which is central to our task. In practice, ground-truth 3D or 4D geometry is even less available than multiview RGB, making rendering-based supervision a practical option for scaling RGB-only data.
>
> ## Weakness 5: Correspondence
>
> The notion of explicit temporal correspondence assumes discrete time steps. In contrast, our model treats time as a continuous dimension unified within the 4D Gaussian representation. Each 4D Gaussian can be interpreted as a local trajectory of a 3D Gaussian over continuous time, meaning temporal coherence is inherently embedded in the representation.
>
> We acknowledge that tracking long-term correspondences—e.g., across complex trajectories—remains beyond the current model's scope, as these often require multiple Gaussians to represent extended motion. Dedicated tracking models like St4RTrack address this, but they are not designed for novel view-time synthesis, which is the focus of our work.
>
> ## Weakness 7 / Formatting Concerns: Literature Review
>
> We placed the related work section toward the end to maintain the narrative flow between the introduction and preliminaries. We are happy to move it earlier if preferred.
>
> The current version was trimmed to meet the page limit, but we have updated it to the best of our knowledge at submission time. We appreciate the reviewer’s feedback and would welcome any specific suggestions on additional works to include. We will revise and expand the related work section accordingly in the camera-ready version.
>
> Below is a compressed version:
>
> - - -
> Prior work on 4D modeling generally falls into three directions: optimization-based, geometry-based, and generation-based, each shaped by different assumptions and data requirements.
>
> - Optimization-based methods reconstruct dynamic scenes by optimizing object- or scene-level representations using multi-view videos [1–4]. While high-quality, they require dense spatiotemporal supervision. To improve generalization and test-time efficiency, recent work introduces depth supervision or lightweight tuning [5,6]. DyST [7] adopts a feedforward Transformer to decompose content, dynamics, and pose from monocular videos, but remains limited in novel view synthesis. Recent approaches explore adapting large reconstruction models (LRMs) to 4D generation [8–10], yet generalizing to arbitrary views and timestamps from sparse inputs remains challenging.
> - Geometry-based methods directly estimate dynamic scene geometry (e.g., depth, flow, camera motion) from videos [11–13]. Inspired by sparse-view geometry for static scenes [14], these approaches extend to dynamics using correspondence-based or monocular depth priors. However, they do not model the full spatiotemporal volume and are unsuitable for novel view/time synthesis.
> - Generation-based methods leverage video diffusion models for 4D synthesis, including explicit asset generation [8,15–17] and controllable video generation [18–20]. These models reduce dependency on multi-view inputs via learned priors but are computationally heavy, prompt-sensitive, and often limited to monocular inputs. Accurate 4D geometry from single-view videos remains fundamentally ill-posed [17].
> Our goal is to develop a generic space-time representation that reconstructs an object from a few views at sparse time points to any view at any time.
>
> [1] Cao, Hexplane: A fast representation for dynamic scenes, CVPR 2022.
>
> [2] Jiang, Consistent4D: Consistent 360° dynamic object generation from monocular video, ICLR 2024.
>
> [3] Yang, Real-time photorealistic dynamic scene representation and rendering with 4D Gaussian splatting, ICLR 2024.
>
> [4] Wang, Shape of motion: 4D reconstruction from a single video, arXiv 2024.
>
> [5] Zhao, Pseudo-generalized dynamic view synthesis from a video, ICLR 2024.
>
> [6] Tian, MonoNeRF: Learning a generalizable dynamic radiance field from monocular videos, ICCV 2023.
>
> [7] Seitzer, DyST: Towards dynamic neural scene representations on real-world videos, ICLR 2024.
>
> [8] Ren, L4GM: Large 4D Gaussian reconstruction model, NeurIPS 2024.
>
> [9] Yang, STORM: Spatio-temporal reconstruction model for large-scale outdoor scenes, ICLR 2025.
>
> [10] Liang, Feed-forward bullet-time reconstruction of dynamic scenes from monocular videos, arXiv 2024.
>
> [11] Feng, ST4RTrack: Simultaneous 4D reconstruction and tracking in the world, arXiv 2025.
>
> [12] Zhang, MonST3R: A simple approach for estimating geometry in the presence of motion, ICLR 2025.
>
> [13] Wang, Continuous 3D perception model with persistent state, CVPR 2025.
>
> [14] Wang, DUSt3R: Geometric 3D vision made easy, CVPR 2024.
>
> [15] Xie, SV4D: Dynamic 3D content generation with multi-frame and multi-view consistency, arXiv 2024.
>
> [16] Yao, SV4D 2.0: Enhancing spatio-temporal consistency in multi-view video diffusion for high-quality 4D generation, arXiv 2025.
>
> [17] Yao, SV4D 2.0: Enhancing spatio-temporal consistency in multi-view video diffusion for high-quality 4D generation, arXiv 2025.
>
> [18] Li, Vivid-Zoo: Multi-view video generation with diffusion model, NeurIPS 2024.
>
> [19] Watson, Controlling space and time with diffusion models, ICLR 2024.
>
> [20] Wu, CAT4D: Create anything in 4D with multi-view video diffusion models, arXiv 2024.

---

> > ### Author Response · Authors · 2025-08-05
> >
> > Dear Reviewer eKZZ,
> >
> > Thank you again for your time and thoughtful review. We’d like to kindly follow up on your impressions of our rebuttal. We worked carefully to address the concerns you raised with concrete updates, and would greatly appreciate any further feedback when convenient.
> >
> > As we clarified, the monocular video setup you found particularly interesting and practical is a special case of our more general any-to-any formulation. While we include it in our evaluation, our method is designed to go beyond that specific scenario. We’re happy to continue the discussion or clarify any aspects of the work or potential future directions.
> >
> > Thank you again for your consideration.
> >
> > Best regards,
> > The Authors

---

> ### Comment · Reviewer_eKZZ · 2025-08-08
> **Clear Reject**
>
> Firstly, the reviewer really appreciates the effort the authors put into the rebuttal and the other reviewers, as well as the AC's valuable time.
>
> After reading the rebuttal and the opinion of the other reviewers, I clearly recommend a rejection with high confidence for this paper.
>
> 1. By posing the camera to DISCRETE time pose combinations, it is for sure a straightforward superset of single or multi-view-synced videos, but is straightforward and mostly useless. Note the examples the authors listed in the rebuttal, most of them are simultaneously captured multi-view settings, not the DISCRETE images. So it’s better to call them multi-view video, which turns out to be a simpler problem because the observation is more sufficient, where reconstruction is easy and trivial. However, as the main claim of this paper is using the discrete time and pose images (e.g., more like the setting in the D-NeRF Blender dataset, one time one pose and suddenly change to a different camera pose in t+1, and this D-NeRF is considered trivial data because sufficient stereo cues enable the smoothness of the 4D representations to converge), this only corresponds to the last point the author mentioned: “Sparse asynchronous recordings (e.g., crowd-sourced phone videos or event capture).” But such an application is not significant. And what’s worse, no evidence in the current material indicates this method can work in such an application setting out of Objaverse, or even on real data.
>
> 2. Not novel because simply changing the input setup of prior art can achieve similar results. When a simple baseline modification, which seems to be common knowledge in the area, can achieve a similar consequence, unless this small modification is unintuitive, then this is definitely trivial, not novel.
>
> 3. 4D Gaussian is novelty? Not true, the 4D Gaussian (4-dim cov) is a contribution to the prior ICLR work, not this paper. Actually, the 4D Gaussian is not widely used in the current community because it lacks correspondence. Any deformation-field-based 3D dynamic Gaussian can also be interpolated by querying continuous time. And to be honest, given the high FPS of the current literature, most feedforward models can be trivially interpolated with simple heuristics.
>
> 4. What is the bottleneck? I’m not saying that the only problem is data, but more than 90% is data. It seems the author also agrees that data is a problem, but if data is a problem, then papers studying modeling should focus more on how to supervise in a more general format. To be honest, the 4D representation in supervised feedforward model training is not a big deal; the performance difference is marginal.
>
> 5. Motion: the authors seem to have some misunderstanding of the question. Yes, delta deformation can be tiny if the FPS is high. But the concern is that the results shown in this paper have relatively small and trivial total movement; i.e., even if you have high FPS, if you roll out to long time, we should observe large movement, but unfortunately, it’s not the case for the current results. Also, this might be because Objaverse itself is mostly occupied by such trivial small motion.
>
> 6. GS is not fundamental. If you rely on the synthetic Objaverse, the correct way is to directly supervise the full geometry. If you do not want to rely on a synthetic dataset (which is not the main claim of the current material as the reviewer feels), then you should focus more on unsupervised training to avoid such a toy Objaverse.
>
> 7. Correspondence: the author has a wrong understanding of correspondence; correspondence is a deformation field, which is continuous. The main issue of this paper’s representation, as well as the original ICLR 4D Gaussian paper, is that the motion is not explicitly represented, and getting the correspondence field is non-trivial in such an Eulerian formulation.
>
> 8. Toy setting: the toy setting means (1) this paper only shows results on synthetic Objaverse, no real data inference results; (2) why it’s hard to get real results? Firstly, there may be some intrinsic difficulty for the Objaverse-trained model to generalize, but the reviewer believes a more serious problem is the authors may find it very hard to get such a discrete time-pose input setting, because people always have continuous capturing of single or multi-view. So this is the toy.
>
> Given all the above concerns not addressed, I recommend a clear reject for the current version. And the modification needed is too significant to match the NeurIPS bar given the current version, so the reviewer recommends the authors carefully study 4D literature and rethink the real 4D problem for a future venue submission.

---

> > ### Author Response · Authors · 2025-08-08
> >
> > Thank you for your follow-up. We appreciate the continued engagement and would like to clarify several factual inaccuracies and misinterpretations present in the latest response.
> >
> > ## Item 1 (The Most Important Factual Misunderstanding)
> >
> > **The reviewer seems to have confused the pre-training objectives versus the tasks evaluated at inference time**. Similar to how language models are pre-trained with next-token prediction but evaluated across diverse downstream tasks, our model is trained using randomly sampled (pose, time) pairs for input and target as a pre-training objective, and then evaluated on a range of reconstruction and generation tasks with varying camera configurations. This is clearly detailed in Section 3.3 (lines 171–180 for reconstruction; lines 186–187 for generation) and further illustrated in Figure 10 of the Appendix.
> >
> > These settings map to the applications mentioned in the previous response:
> > - Alternating Canonical Views / Frame Interpolation → few fixed cameras (e.g., surveillance or mocap setups);
> > - Two Rotating Cameras → continuously moving cameras (e.g., mobile robotics, sports broadcasting);
> > - Random Input Views → sparse asynchronous recordings (e.g., crowd-sourced phone videos);
> > - 4D Asset Generation from Monocular Video → matches the setting in SV4D and L4GM.
> >
> > We would like to respectfully clarify that the reviewer’s claim that “this only corresponds to the last point the author mentioned” is factually incorrect. The configurations we evaluate are not limited to the asynchronous case; they encompass a wide variety of spatial and temporal sparsity patterns, **including the monocular to multiview setup the reviewer themselves describes as “a more real and practical problem” in the initial review**. As such, this setting of L4GM (NeurIPS 2024) is just a subset of our evaluation.
> >
> > ## Item 2
> >
> > If the “prior art” refers to GS-LRM, we explicitly show in Figure 3 and Table 5 that simply interpolating 3D-LRM does not match the performance of 4D-LRM as information sharing across frames is essential. If it refers to L4GM, our model is architecturally distinct: L4GM is autoregressive. With the same multiview prior, we achieve better performance, as shown in Figure 6. If the reviewer believes that “a simple baseline modification” can achieve similar results and that this is “common knowledge,” we respectfully ask for a specific citation. To our knowledge, no prior work demonstrates these capabilities.
> >
> > ## Item 3
> >
> > If the reviewer believes that deformation-based dynamic Gaussians can be queried over continuous time, likewise, 4D Gaussians with 4D covariance can be sampled along the time dimension to approximate deformation fields—both are valid formulations with tradeoffs. We also wish to note that it is inappropriate to disparage the previous work as part of a review.
> >
> > ## Item 4
> >
> > This review contains several broad claims about the field (e.g., “90% is data,” “representation doesn’t matter,” “performance is marginal”) that are unsupported and not grounded in existing literature. If such statements are to be taken seriously in a peer review, they must be backed by concrete evidence or citations. Our experiments (Figure 8, lines 226-227) clearly show that representation and modeling choices significantly affect performance.
> >
> > ## Real Data
> >
> > We appreciate the suggestion and agree that real-world testing is important. To address this, we adapted the Plenoptic Video dataset, which contains human motion in real indoor scenes. We select 6 scenes and preprocess them via center cropping and downsampling to 256×256 resolution with FOV 50 to match our training settings. For each sequence, we sample 24 frames across time and render to 6 novel views. We evaluate under two input settings: using raw frames vs. using SAM2 to segment the dynamic human.
> >
> > | Input Type | PSNR  | LPIPS | SSIM |
> > |------------|-------|-------|------|
> > | Raw        | 24.73 | 0.06  | 0.95 |
> > | Segment    | 27.53 | 0.03  | 0.98 |
> >
> > While there is a performance drop compared to pure object-centric data (due to domain shift and background clutter), the results are non-trivial and demonstrate generalization, especially considering the small number of input views (24) without optimization.
> >
> > We note that Objaverse already includes real scene-normalized data, and prior work (e.g., GSLRM) has shown that object-level training can generalize to simple scene-level layouts. While we have not trained a full scene-level 4D-LRM due to the lack of license-permissive 4D scene datasets, we view this as promising future work. The underlying design of 4D-LRM is extensible to more complex real-world scenarios.

---

> > > ### Author Response · Authors · 2025-08-08
> > >
> > > ## Items 5-8
> > >
> > > We would like to address the review’s critique regarding our formulation and its assumptions. Several claims in this critique are stated as objective facts but are presented without empirical justification or citation. These include, for example: “Objaverse itself is mostly occupied by such trivial small motion,” “the correct way is to directly supervise the full geometry,” “you should focus more on unsupervised training to avoid such a toy Objaverse,” “correspondence is a deformation field, which is continuous,” and “the motion is not explicitly represented… in such an Eulerian formulation.” We believe such assertions should be supported by evidence or literature references, especially when used to critique a valid and peer-reviewed approach.
> > >
> > > Beyond being unsubstantiated, we respectfully note that this critique also dismisses an established and published formulation (Eulerian, unified 4D Gaussian representations) based solely on their preference for correspondence-based / Lagrangian modeling. While we acknowledge and value the contributions of deformation-field-based approaches, it is inappropriate to assume that they represent the only “correct” paradigm for 4D modeling. There is no evidence that Eulerian representations are inherently flawed, and importantly, **we already show evidence in Figure 8 that the deformation-field-based 4D Gaussian (CVPR 2024) underperformed** in our experiments.

---

### Official Review · Reviewer_3UsT · 2025-07-02

**Clarity:** 3
**Significance:** 2
**Originality:** 3
**Rating:** 4
**Confidence:** 4

**Summary:**

This paper introduces a 4D reconstruction approach, which is able to recover s sequence of 3D-GS from a set of sparse views in temporal or spatial dimension. In general, the task setting is interesting and the visualized results on synthetic data is good. Also, the method is clearly presented and the whole architecture is not very surprising that it can work on current task setting. The major concern is about the generalization ability to real world. Previously, people show real world 3D reconstruction from single RGB image (such as LRM), where readers can clearly expect the reconstruction ability. However, authors didn't show any real-world results in current manuscript, and all used examples look very artificial and very similar to training data in Objaverse-4D. Hence, i am concerning if the proposed method can really generalize to real-world, which is more useful and more convincing (after using 160 GPUs). For now i suggest a borderline rejection, and i will consider to increase score if authors show more convincing results in rebuttal.

**Questions:**

My main question is regarding the promised code release of the manuscript. Authors don't have any claim in the abstract like normal papers, but promise the code and data released in the 5. Open access to data and code in the checklist. Could authors confirm if the original training / inference / data in the submission will be released? If yes, could you please add in the abstract of the revised version?

As mentioned in the summary and weakness sections, if authors can show real world 4D reconstruction with scenarios like moving human / moving pets reconstruction, the proposed methods will be more convincing, and i will increase the score.

**Ethical Concerns:**

["NO or VERY MINOR ethics concerns only"]

**Final Justification:**

After authors explanation in the rebuttal. My primacy concern is addressed. I appreciate authors for promising the code release hope authors can include the additional visualization in the final version. I am also curious to see the future works regarding changing to pose-free input video setting.

**Limitations:**

Yes, limitations are discussed in the main paper.

**Paper Formatting Concerns:**

The paper formatting looks good.

**Quality:**

3

**Strengths And Weaknesses:**

- The manuscript has very clear writing which is easy to follow. Also the presented figures are intuitive and easy to understand.
- As mentioned in the summary, authors didn't present any real world objects. The proposed task is interesting and attractive for such a real-world scenario: if i capture several photos of my dog from different perspective and it is moving during the capture, how does the method perform? I think it should still be feasible to perform since authors show the goldfish 4D reconstruction and i assume there are other animals in the training data. If authors can provide several of these results, i will consider to increase score.
- Another application could be reconstructing moving human. Authors could leverage multi-view human performance capture data for training / evaluation.
- The method requires camera poses for the input views. This is a big limitation as estimating camera pose from few-views captures of deforming objects is a very complex and unsolved problem. Authors should consider how to address this problem, and i will consider to increase score from boarder line to accept / strong accept.

---

> ### Author Rebuttal · Authors · 2025-07-31
>
> We thank the reviewer for the detailed and constructive feedback. We are glad that the task setting was found to be "interesting" and that the visualized results were "good." We appreciate the recognition that our method is "clearly presented."
>
> ## Question 1: Code Release
> We confirm that a cleaned version of the training and inference code, along with data, will be released upon publication. We will also update the abstract in the revised version to explicitly reflect this release.
>
> ## Weakness 1 / Question 2: Real Objects
>
> We appreciate the suggestion and agree that real-world testing is important. To address this, we adapted the Plenoptic Video dataset, which contains 6 human motions in real indoor scenes. We preprocess them via center cropping and downsampling to 256×256 resolution with FOV 50 to match our training settings. For each sequence, we sample 24 frames across time and render to 6 novel views. We evaluate under two input settings: using raw frames vs. using SAM2 to segment the dynamic human.
>
> | Input Type | PSNR  | LPIPS | SSIM |
> |------------|-------|-------|------|
> | Raw        | 24.73 | 0.06  | 0.95 |
> | Segment    | 27.53 | 0.03  | 0.98 |
>
> While there is a performance drop compared to pure object-centric data (due to domain shift and background clutter), the results are non-trivial and demonstrate generalization, especially considering the small number of input views (24) without optimization over hundreds of posed images. Also, scene-level LRMs usually adopt different normalization strategies from the start.
>
> We note that Objaverse already includes real scene-normalized data, and prior work (e.g., GSLRM) has shown that object-level training can generalize to simple scene-level layouts. While we have not trained a full scene-level 4D-LRM due to the lack of license-permissive 4D scene datasets, we view this as promising future work. The underlying design of 4D-LRM is extensible to more complex real-world scenarios.
>
> ## Weakness 2: Camera Poses
> We use posed RGB images as input because our primary goal is to investigate what 4D representation and training objectives enable scaling LRM-style reconstruction models. Thus, camera poses are a controlled variable, not a limitation of the concept. That said, adapting our model to work without known poses is a promising next direction. Recent methods like PF-LRM and RayZer demonstrate how to jointly predict camera poses and scene representation from unposed, uncalibrated multi-view inputs, using RGB-only supervision:
>
> - PF-LRM [1] predicts coarse geometry and solves for pose using a differentiable PnP module combined with self-attention across object and image tokens.
> - RayZer [2] recovers both camera parameters and scene representations end-to-end, without any ground-truth pose or geometry, using a transformer-based ray-aware architecture.
>
> We believe pose-free approaches developed for 3D can transfer to the 4D setting, as even when part of the object deforms, static or rigid regions can serve as anchors for relative pose prediction and joint optimization. Due to the limited time in the rebuttal phase, we are unable to train such a model from scratch. Yet we expect that integrating such techniques into a pose-free variant of our 4D-LRM pipeline is feasible and would be a natural extension toward making the approach applicable to real-world RGB-only captures.
>
> [1] PF-LRM: Pose-Free Large Reconstruction Model for Joint Pose and Shape Prediction. ICLR 2024.
>
> [2] RayZer: A Self-supervised Large View Synthesis Model. ICCV 2025.

---

> > ### Comment · Reviewer_3UsT · 2025-08-05
> >
> > I appreciate authors for providing the rebuttal. Since authors promise the code and the data release, i am generally positive to the submission. Regarding the real objects experiments, it's a pity that no visualization can be included in the rebuttal. I hope authors will also include this in the final submission because it helps to understand the generalization gap from synthetic training to real inference. Regarding the camera poses, i think predict the camera poses of input video frames should be do-able and it can be the direction for follow-up works.

---

> > > ### Author Response · Authors · 2025-08-05
> > >
> > > Thank you for your thoughtful review and for being generally positive toward the submission. We’re glad to hear that your concerns have been addressed.
> > >
> > > Just a gentle reminder: if you feel the revision merits an update, you may use `Edit > Review Revision` to adjust your scores accordingly when you get a chance to. We sincerely appreciate your time and feedback for our work!

---

> > > ### Comment · Area_Chair_7XhB · 2025-08-08
> > > **Please remember to update your final rating**
> > >
> > > Hi, Reviewer 3UsT
> > >
> > > Thanks for your efforts to make the responses to the rebuttal. It seems you are change to be positive after the rebuttal. However, I found you final rating is still "borderline reject". So, could you please update it accordingly？
> > >
> > > Thanks,
> > > AC

---

### Official Review · Reviewer_NLGT · 2025-07-02

**Clarity:** 3
**Significance:** 3
**Originality:** 3
**Rating:** 5
**Confidence:** 3

**Summary:**

The paper proposes a novel, any-to-any formulation of the temporal scene reconstruction task, where given unconstrained posed views from different timestamps, the task is to render arbitrary novel view-time combinations (within the range of input space and time). The paper then introduces a large scale reconstruction model for this task. Importantly the paper shows that pre-training on the any-to-any reconstruction task and then fine-tuning for the 4D generation task, the method improves SOTA in an established benchmark by a large margin.

**Questions:**

-  Have you considered using latent gaussians, similar to e.g. in [70], to address the resolution issue?

**Ethical Concerns:**

["NO or VERY MINOR ethics concerns only"]

**Final Justification:**

All my concerns have been addressed. I reviewed the concerns raised by other reviewers, and i believe they have been mostly addressed adequately. I raised my score accordingly.

**Limitations:**

Limitations are explicitly discussed.

**Paper Formatting Concerns:**

no concern

**Quality:**

3

**Strengths And Weaknesses:**

Strength

-	The problem formulation of any-to-any space-time reconstruction is an interesting, and useful generalization of prior problem formulations.

-	The proposed method is reasonable, the design choices are well motivated and explained.

-	The demonstrated results, including on the Constistent4D benchmark, are impressive, although some aspects need to be clarified (see weaknesses).

-	The evaluation is thorough and informative, including scalability with data.

-	The paper is generally well written and easy to follow.

-	I appreciated the limitations section.

Weaknesses

-	Some choices for the experimental setup need to be clarified. Line 150 mentions “we hold out 56 challenging objects with more complex motion”. What was the motivation for removing challenging cases? It makes me wonder whether the results were less favorable on these examples. It would be important to provide a strong justification, or include results on the full benchmark dataset, even if they are not as strong.

-	Training and rendering times should be reported along rendering quality metrics.

-	The compute requirements could be more clearly communicated in the main text, especially in Sect 4.2 it would be good to specific training times explicitly.

I am willing to raise my score if the above weaknesses can be addressed.

---

> ### Author Rebuttal · Authors · 2025-07-31
>
> We thank the reviewer for the thoughtful and constructive feedback. We are encouraged that the reviewer found the problem formulation of "any-to-any space-time reconstruction" to be an "interesting, and useful generalization of prior problem formulations," and appreciated that "the proposed method is reasonable" and that "the design choices are well motivated and explained." We are also glad that the "evaluation is thorough and informative, including scalability with data."
> ## Weakness 1: Misunderstanding about Challenging Cases
>
> Sorry for the misunderstanding, and we apologize for the ambiguity in the wording. To clarify: the 56 challenging objects with more complex motion were **not excluded** from evaluation. Rather, they were **explicitly held out from pre-training** to serve as an extended test set for further evaluation to **avoid data leakage**. They are in addition to the re-rendered Consistent4D test set, which is less challenging compared to this test set. We agree this could be misread as exclusion and will revise the sentence for clarity.
>
> ## Weakness 2: Training & Rendering Time
>
> We agree that reporting training and inference efficiency alongside rendering quality is important. Due to space constraints, we briefly described runtime in Section 3.2 and included full details in the Appendix, but will surface this information more clearly in the camera-ready version. For clarity:
>
> - Training: 4D-LRM is trained in two stages on 160 A100 (40G VRAM) GPUs. Pretraining (128×128) takes ~5 days, and fine-tuning (256×256) takes an additional ~5 days. For 4D asset generation, fine-tuning with free Gaussians takes ~16K steps on 64 A100 GPUs.
> - Inference: Rendering a 24-frame sequence (which is the case for the main experiments) with 24 input views takes ~1.25s for 4D-LRM-Large and ~1.90s with free Gaussians, measured on a single A100 GPU averaged over 25 trials.
>
> We will make these numbers more prominent in the main text.
>
> ## Weakness 3: Training Time For Scaling Analysis
>
> We appreciate the suggestion. In Section 4.2, we report training steps (rather than time) on the x-axis for scaling analysis as it provides a consistent and implementation-independent measure of progress, which is standard practice in prior work. Wall-clock time is less reliable due to fluctuations caused by memory fragmentation in mixed-precision buffers, which accumulate over time. For example, during 128×128 pretraining, the step time increases from ~3s initially to ~5.25s later, leading to a total of ~113 hours for 100K steps. While mitigations like periodic restarts exist, they introduce additional variability. For this reason, we report total training time (e.g., 5 days for each stage) in Section 3.2, and will clarify this rationale in the revised text.
>
> ## Question: Latent Gaussian in STORM
>
> Thank you for the thoughtful suggestion. We appreciate the reference to [70] (STORM) and the discussion around latent Gaussians.
>
> We note that in Table 1 of [70], the latent Gaussian variant is slightly less performant than the full STORM model on the general Waymo Open Dataset. As the authors also acknowledge, the latent design is primarily introduced to improve performance in settings with **large novel-view extrapolation** or in **domain-specific applications** like human body modeling.
>
> To empirically test this idea in our setting, we conducted a small-scale comparison by integrating the **patch-aligned Gaussian mechanism** (central to latent Gaussians in [70]) into **4D-LRM-base**, using a similar setup as Section 4.2. After 40K training steps, we observed:
>
> | Gaussian Alignment | PSNR   | LPIPS | SSIM  |
> | ------------------ | ------ | ----- | ----- |
> | Pixel (Ours)       | 25.913 | 0.114 | 0.867 |
> | Patch ([70])      | 24.135 | 0.154 | 0.832 |
>
> These results suggest that our pixel-aligned Gaussians offer better fidelity in our reconstruction task. This also echoes the findings of Table 1 in [70]. That said, the concept of using a latent Gaussian backbone is promising, especially for **extrapolation-heavy or low-coverage regimes**. We consider this an exciting direction for future work.

---

> > ### Comment · Reviewer_NLGT · 2025-08-05
> >
> > Thank you for the careful response and additional results. All my concerns have been addressed, and i will raise my score accordingly.

---

> > > ### Author Response · Authors · 2025-08-05
> > >
> > > Thank you for your thoughtful review, for raising the score, and for confirming that all your concerns have been addressed!

---

### Note · Authors · 2025-08-15

The authors would love to sincerely thank all reviewers and the Area Chair for their constructive feedback and insightful discussions, which have greatly helped us clarify and strengthen our work!

## Strengths and Contributions

- **Flexible Any-to-Any 4D Reconstruction Framework**: 4D-LRM adopts a general formulation of 4D reconstruction that supports arbitrary input/output configurations, which is "interesting, and useful generalization" of prior 4D formulations (`NLGT`) and the "M-in-N-out" 4D setup with "limited assumptions on the input" (`DFse`).
- **Principled Design with Generative Capacity**: The "design choices are well motivated and explained" (`NLGT`), the use of "optional learnable free Gaussian tokens" that "work surprisingly well" (`DFse`), and we demonstrate "scaling behavior... despite... limited data source" (`eKZZ`).
- **Thorough and Multi-Scale Evaluation**: Our "evaluation is thorough and informative" (`NLGT`), "extensively evaluated... under different test settings" (`DFse`).

## Addressed Concerns
We’ve carefully addressed all concerns raised by the reviewers and are grateful that most feedback has been constructive and engaging.

- **Additional Comparisons**: We have included additional comparisons to Latent Gaussian in STORM (`NLGT`) and to Optimization-based 4D Reconstruction (`DFse`).
- **Generalization to Real Objects**: We have adapted the Plenoptic Video dataset for additional evaluation (`3UsT, eKZZ, DFse`).
- **Conceptual Clarifications**:
   * **Camera Poses**: We have discussed next steps on how pose-free versions of 4D-LRM can be developed, and Review `3UsT` agreed that "predicting the camera poses of input video frames should be doable and it can be the direction for follow-up works."
   * **Unrealistic Problem Setup**: Any-to-any is the pre-training objective of 4D-LRM, not how it is supposed to be evaluated. We have clarified to Reviewer `eKZZ` on the pre-training objectives with the tasks evaluated at inference time. A wide variety of 4D task formulations can be handled at test time, including the monocular to multiview setup, which the Reviewer `eKZZ` describes as “a more real and practical problem.”

---

### Decision · Program_Chairs · 2025-09-17

**Decision:**

Accept (poster)

**Comment:**

The paper received mixed ratings initially. After the rebuttal, three reviewers are positive and only one remains negative. During the discussion period, the negative reviewer raised many issues which also received very detailed responses from authors. The AC carefully read all responses and think the issues are well addressed. Regarding the negative reviewer did not reply again, the AC finally recommend acceptance.